# Acceptance of Smart Electronic Monitoring at Work as a Result of a Privacy Calculus Decision

**Evgenia Princi * and Nicole C. Krämer**

Department of Computer Science and Applied Cognitive Science, Social Psychology: Media & Communication, University of Duisburg-Essen, 47057 Duisburg, Germany

* Correspondence: evgenia.princi@uni-due.de

**Abstract:** Smart technology in the area of the Internet of Things (IoT) that extensively gathers user data in order to provide full functioning has become ubiquitous in our everyday life. At the workplace, individual's privacy is especially threatened by the deployment of smart monitoring technology due to unbalanced power relations. In this work we argue that employees' acceptance of smart monitoring systems can be predicted based on privacy calculus considerations and trust. Therefore, in an online experiment ($N = 661$) we examined employees' acceptance of a smart emergency detection system, depending on the rescue value of the system and whether the system's tracking is privacy-invading or privacy-preserving. We hypothesized that trust in the employer, perceived benefits and risks serve as predictors of system acceptance. Moreover, the moderating effect of privacy concerns is analyzed.

**Keywords:** IoT technology; electronic monitoring at work; privacy calculus; privacy concerns; smart technology; trust

## 1. Introduction

The rapidly increasing use of Internet of Things (IoT) technology accompanies our everyday life, simplifying and accelerating diverse processes. IoT devices, also called 'smart technology', are able to interact with people, with other devices and with their environment, exchanging information with intelligent algorithms enabling automated decision-making. In order to provide full functioning, smart technology extensively gathers data, using various sensors and different tracking methods [1]. The collected information is processed and analyzed automatically, applying algorithms to specific user requirements, such as individually tailored services and personalized content [2,3]. As a result, privacy has become established as a tradable good with user data as the means of payment.

The number of IoT devices is tremendously growing in private households (tablets, wearables, smart household appliances etc.). In industry, smart technology (smart buildings, process automation etc.) is deployed, amongst other things, for staff security (e.g., smart emergency detection systems) or for competitive reasons (e.g., protection from corporate espionage). However, applying smart technology for the purpose of electronic monitoring in the workplace means a profound invasion of privacy [4], particularly as power relations are unbalanced between management and staff, as employees usually do not have decisional power regarding the deployment of IoT technology, the intended purpose of the monitoring, the kind of information to be tracked, its storage and utilization. Moreover, if people consider privacy risks of a smart monitoring system as prevailing, their possible actions at the workplace are limited. Employees could either evade being tracked by quitting their job or condone the system's deployment with stress and discomfort as possible consequences. Accordingly, the employer's decision to deploy smart technology might lead to mistrust and a lower commitment on the part of the staff potentially reducing productivity. Therefore, it is essential to understand how employees perceive smart monitoring technology capable of data tracking and what factors contribute

to their acceptance of the system. In order to determine the mechanisms behind the evaluation of smart technology which might be a threat to individual's privacy, we take privacy calculus [5] and communication privacy management (CPM) [6] as a theoretical background. The current study aims to re-examine privacy calculus, evaluating its applicability within the framework of smart technology usage at the workplace where possibilities for employees to react are limited, and to discuss whether the amount of tracking can impact CPM. Furthermore, we investigate how employees' privacy concerns and trust in the employer are related to the acceptance of the IoT system.

## 2. Literature Review

### 2.1. Trust in the Employer

As the relationship between the employer and staff is marked by an imbalance regarding management processes such as decisional power or control over information, trust is a crucial factor when it comes to predispositions of employees towards being monitored [7]. Trust can be understood "as the confidence that the other party to an exchange will not exploit one's vulnerabilities" [8] (p. 2). Considering the amount of available personal data an IoT system is capable of tracking, the deployment of such a technology might threaten employees' privacy. Research in the context of self-disclosure online could show that trust correlates with disclosing behavior as people are likely to provide information when they trust the company asking for it [9] and that trust plays a key role regarding the decision whether to disclose personal information on social networking sites and other websites [10]. However, in a work-related context, employees do not actively disclose their information. Rather, their data is collected on a large scale when the employer decides to deploy a smart monitoring system capable of data tracking. While Jøsang et al. [11] posited that the willingness of individuals to provide private information is higher when they have a trusting relationship with their employer, other scholars demonstrated that intense employee monitoring can damage trust resulting in a less efficient performance [4], and that the perceived amount of data tracking at the workplace negatively affects commitment and trust [12]. In their study on the intention to use wearable devices at the workplace, Yildirim and Ali-Eldin [13] demonstrated that employees with a high level of privacy concerns in terms of data collection and improper access showed little trust in the employer. Therefore, in this work, we argue that trust in the employer is a fundamental construct when it comes to the acceptance of a smart monitoring system deployed at the workplace. In particular, we assume that employees who show a trusting relationship with the employer will also have more trust regarding the handling of their personal data and thus have a higher acceptance of smart technology that is able of data tracking. Furthermore, academic literature references trust as a pivotal construct in situations with uncertainty and fear as trust facilitates to overcome perceived risks [14,15]. Thus, we expect that employees will perceive less privacy risks from an IoT system when they trust their employer. We therefore formulate the following hypotheses:

**Hypothesis 1 (H1):** *There will be a positive relationship between employees' trust in the company and their acceptance of a smart emergency detection system.*

**Hypothesis 2 (H2):** *There will be a negative relationship between trust and perceived risks.*

### 2.2. Privacy Calculus in the Framework of Smart Technology

In recent years, many attempts have been made to examine how people negotiate their privacy in exchange for certain benefits and what factors have an impact on this trade [2,10,16]. However, within the framework of privacy calculus theory [5] describing the trade-off between expected benefits and anticipated risks of providing personal data, research focuses on individual's self-disclosure, usually in the context of social networks (e.g., [17]) or throughout different websites (e.g., [10]). Accordingly, academic literature investigating privacy calculus as the theoretical basis for the utilization

of IoT technology has been limited. Adapting privacy calculus to this field calls people's bargaining position into question, as smart devices are able to collect large amounts of personal information automatically without people actively disclosing it [18].

With regard to technology acceptance, many theories have been proposed by scholars. The technology acceptance model (TAM; [19]) found substantial and empirical support and is well-studied in this field of research. Particularly, it aims at predicting a person's behavioral intention to use a system as well as actual usage depending on perceived usefulness (i.e., to what extent the system helps enhancing individual's job performance) and perceived ease of use (i.e., the required effort of the usage). A theoretical advancement of TAM and other constructs is the unified theory of acceptance and use (UTAUT; [20]). This extensive model takes additional factors, such as social influence and voluntariness of use, as determinants of usage intention and behavior into consideration. At this point, it becomes clear that theories and models of technology acceptance have a common goal, which is to explain behavioral intention and usage adoption. However, since employees are affected by the system, but apart from being monitored they do not engage in activities with it, they do not have the choice to adopt or reject the system but rather to evaluate to what degree they approve the authority decision regarding the system's implementation at the workplace. Accordingly, most constructs taken up by technology acceptance models, such as investigating the effect of system usage on job performance, are not applicable in this context. Furthermore, in his diffusion of innovations theory, Rogers [21] described innovation diffusion as an adaptation process where individuals first learn about the innovation's existence, form an attitude towards it and engage with it before finally putting it into use. However, specifically in the context of organizations, an "authority innovation-decision is one with which the organization's employees must comply" (p. 403). In other words, in the case of the implementation of an IoT monitoring system at the workplace, employees have no decisional power, which is why common technology acceptance models seem less suitable than the risk-benefit trade-off postulated by the privacy calculus. Finally, these models do not consider the privacy aspect which plays a decisive role for the acceptance of tracking systems, given their potentially privacy intrusive character. In the light of the above, privacy calculus serves as the main theoretical foundation investigating system acceptance as an outcome of a risk-benefit trade-off.

*2.3. Privacy Challenges of Smart Monitoring at the Workplace*

In order to raise an organization's efficiency, workplace monitoring has become a convenient instrument for employers, promising positive outcomes such as a higher security of employees (e.g., by identifying potential safety hazards) or support of HR decision-making by providing insights about job performance. Recent literature provides an overview of different approaches regarding the development of privacy-preserving IoT technology. Studies demonstrate technical possibilities to reduce privacy risks, for example by providing control over data to be collected [22], authorization restriction [23] or continuous anonymization [24]. However, numerous IoT systems still have no privacy protecting measures leading to enormous challenges for employee privacy as a result of the vast collection of personal data. Intelligent IoT tracking systems are able to gather masses of sensitive data, correlating it to employee performance and automatically drawing conclusions on improvable aspects regarding efficiency, which is called people analytics [25]. From the legal point of view, employees give up a lot of privacy expectations since "The employer is allowed to monitor employees through supervisors, video cameras, computer software, or other methods that capture employees working within the scope of employment." [26]. However, there are limits to employee surveillance, for example if monitoring is undisclosed. Secret tracking requires severe and legitimate reasons, such as suspicion of fraud or theft [26]. Generally, employees should be provided with information about being subject of data collection (e.g., by information signs). Accordingly, the commencement of the European General Data Protection Regulation (GDPR) guarantees individuals the right to be informed about the processing purposes, the legal basis or automated decision-making processes. It should be noted, however, that despite of the GDPR being an important step in the direction of elucidated

individuals the reality looks different. Privacy policies might be difficult to understand in terms of technical terminology and lack transparency [27,28]. Additionally, employees might underestimate the privacy-risks of the technology implemented at the workplace or simply not care about the provided information leading to a lack of awareness and knowledgeability. Furthermore, legal regulations between countries have to be differentiated [29].

Within the theoretical framework of electronic monitoring at work, Watkins Allen et al. [4] investigated the CPM theory [6]. The five principles of CPM state that individuals believe to (1) have a right to control their private information, (2) that they decide whether to disclose private information based on personal privacy rules, (3) that people with access to someone's private information become co-owners of that information, (4) that they have to negotiate new privacy rules regarding information disclosure and (5) that this must lead to privacy boundary turbulences when the rules are unclear or not mutually agreeable. Boundary turbulences means that a violation of an individual's privacy rules results in a conflict regarding the tension between disclosure and concealment of private information [30].

According to CPM, smart electronic monitoring at work might comprise the privacy of employees by accessing private information notwithstanding individual privacy rules. Moreover, the number of co-owners is incalculable and there is no space left for negotiations of one's privacy increasing the likelihood of boundary turbulences. This goes in line with the information limit postulated by Sutanto et al. [31], which showed parallels to CPM regarding individual's possibility to determine what information to disclose. Sutanto et al. [31] posited that people accept a certain amount of their private information being collected. However, when this limit is reached, the willingness to accept data tracking declines. Similar findings were reported by Chang et al. [12] who found that the amount of perceived monitoring negatively influences employees' commitment. Therefore, it is crucial to consider whether the IoT system is privacy-preserving or privacy-invading. Privacy-preserving IoT technology can be developed when privacy is implemented by design. This means that technology is shaped in a way that enables data protection for example by minimizing or avoiding collection of personal data, which is also required by article 25 of the GDPR (eugdpr.org). Possible approaches for data protection are local data storage or transparency regarding the purpose of data collection [32]. Privacy-invading technology, on the other hand, can comprise the identification of individuals and unlimited collection of personal data [33]. Notwithstanding the conformity with the GDPR, as in the workplace employees are often exposed to a frequent monitoring [34] their information limit in the long term might be exceeded leading to a decreasing acceptance of privacy-invading IoT technology. This corroborates the findings from Porambage et al. [33], stating that individuals will rather accept privacy-preserving IoT deployments.

Nevertheless, people still tolerate collection of their information in exchange for an anticipated positive outcome (e.g., higher security ensured by a smart safeguard system), overweighing perceived risks. Previous studies showed that certain benefits people expect from a product or service change their behavior regarding the disclosure of private information. Users share more data on social networking sites (e.g., XING) for their free use [35] or when expecting other benefits, such as maintaining relationships [36]. When it comes to the deployment of IoT technology, Zheng et al. [37] confirmed that people were largely using smart devices in their homes for the reason of convenience promised by the technology. Correspondingly, perceived advantages have to outweigh perceived privacy risks in order for people to provide personal data. The deployment of a smart monitoring system at the workplace can also provide additional value for employees such as security [36] by automating and accelerating emergency recognition. Furthermore, employees in a poll by Watkins Allen et al. [4] reported monitoring as beneficial or necessary. Therefore, we can assume that a high rescue value of a smart monitoring system will be perceived as a pertinent benefit exceeding perceived risks and leading to a higher acceptance of the system.

In the current study, we hypothesized that employees will weigh the system's privacy risks resulting from its tracking ability against the benefits regarding its efficiency in terms of the system's

rescue value. Furthermore, we expected that perceived benefits of the IoT system will be positively related to system's acceptance while the perceived risks will be negatively related to it.

As posited by Romanou [32], people are aware of risks emerging from smart technology (e.g., malicious behavior or data theft), which makes them more suspicious towards data tracking. Therefore, we assume that the more data is tracked by the IoT system, the more privacy risks people will perceive. Based on the previous notions, the following hypotheses were derived:

**Hypothesis 3a (H3a):** *The amount of tracking will have a negative effect on employees' acceptance of the system.*

**Hypothesis 3b (H3b):** *The amount of tracking will have a positive effect on perceived risks.*

**Hypothesis 4 (H4):** *The rescue value of the smart emergency detection system will have a positive effect on employees' acceptance of the system.*

**Hypothesis 5a (H5a):** *There will be a positive relationship between perceived benefits and the acceptance of a smart emergency detection system.*

**Hypothesis 5b (H5b):** *There will be a negative relationship between perceived risks and the acceptance of a smart emergency detection system.*

**Hypothesis 6 (H6):** *Perceived benefits will be negatively related to perceived risks.*

*2.4. Privacy Concerns and the Value of Privacy*

Given the permanent tracking automation of smart technologies, organizational surveillance by some scholars is referred to as the panopticon metaphor (e.g., [34]) illustrating the employer's superiority regarding the control of personal information. Moreover, as stated previously, considering the sensitivity of collected data [38] individual's information limit [31] is quickly exceeded and the right of data ownership an individual perceives, according to CPM, might be severely violated meaning a privacy threat for employees. The occurring worries about privacy can cause negative emotions such as privacy concerns [39,40]. A number of studies showed that people generally tend to be concerned about their privacy (e.g., [41]) and that this tendency is affected by an individual's risk perceptions [42] or might for its part influence risk beliefs [43]. This fact might be traced back to what has been described as a universal and cross-cultural need for privacy [44,45] leading individuals to strongly valuing their privacy [46]. With this regards, Trepte and Masur [47] demonstrated people's demand to determine what information should be publicly available and the belief that the protection of privacy should be enshrined in the constitution, which the authors referred to as the value of privacy. However, examining the value of privacy, Krasnova et al. [48] found that individuals have a varying degree of subjectively perceived privacy concerns and, accordingly attach a different value to their privacy. This might have an effect on the extent to which individuals perceive the data tracking as a threat to their privacy and thereby moderate the relationship between the amount of tracking and the acceptance of the IoT system. This corroborates other studies investigating the moderating effect of privacy concerns [49–51]. Yun et al. [51] demonstrated that the effect of performance expectancy on continuous usage intention of location-based services is stronger when people have low privacy concerns. Tan et al. [50] found privacy concerns to be a moderator in the TAM. To be more precise, they demonstrated that the effect of perceived usefulness and perceived ease of use on users' intention to continue to use social networking sites varies with different levels of privacy concerns. Considering that people show reluctant attitudes when asked for too much personal information [39,40] we assume that individuals who have higher privacy concerns will rather avoid IoT technology with a high amount of tracking in order to protect their privacy. Furthermore, as prior studies showed an interdependence between privacy concerns and perceived risks [42,43] we expected that the privacy risks, which people see in the deployment of IoT technology will be higher when privacy concerns are high.

**Hypothesis 7 (H7):** *The effect of tracking on the acceptance of a smart emergency detection system will be stronger among people with high privacy concerns.*

**Hypothesis 8 (H8):** *There will be a positive relationship between privacy concerns and perceived risks.*

To examine the hypotheses, we conducted an online-study, explained in more detail in the following sections. The model can be seen in Figure 1.

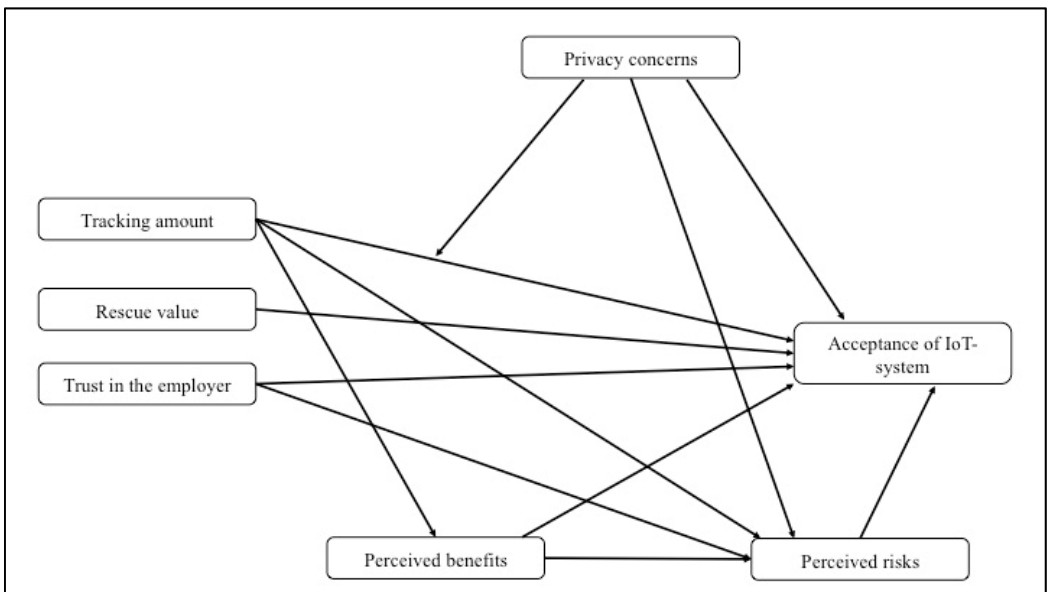

**Figure 1.** Path model of the investigated constructs.

## 3. Materials and Methods

### 3.1. Sample and Design

As the focus of the study is on electronic monitoring at the workplace, the study was directed at employees in order to include participants who are familiar with the context. Participants were recruited online via Facebook groups and survey distribution platforms (e.g., www.surveycircle.com) and had the chance to win a gift card worth 5–100 €. A total of 770 participants took part in the online-survey. In a first step, we filtered out participants who were underage, failed to respond to the control questions and those who showed unreasonable reading times, which we tested to be less that 220 s, resulting in an overall sample size of 661 individuals. The sample included 206 males, 447 females and eight individuals who did not specify their gender with an age range of 18 to 65 ($M = 27.67$, $SD = 7.51$).

We employed a $2 \times 2$ between-subjects design, manipulating the amount of tracking (see Table 1) and the rescue value of the IoT system (see Table 2). Each of the four conditions included a different vignette describing a smart emergency detection system with a high/ low rescue value and privacy-preserving or privacy-invading tracking capability. Due to the high level of heterogeneity of IoT devices and the diverse fields of their implementation, an assessment of the perception of this technology in general proves to be rather difficult. In terms of privacy, an individual might distinguish between technology used at home and technology used by the employer. Furthermore, in regard of potential privacy threat, the evaluation of IoT systems might depend on the respective device. It is conceivable that a smart freezer will raise less privacy concerns than for example an intelligent speaker, like Amazon's Alexa. Therefore, users' perception of a specific system, namely an IoT monitoring system, is assessed individually in this study. In order to present realistic versions of a smart emergency detection system, the different features as well as their description were based on

information from manufacturer websites (e.g., www.rhebo.com). Thus, tracking functions were named without particularizing which sensors are implemented, where collected data are stored or how they are processed. The detailed scenarios can be found in Appendix A.

Participants were randomly assigned to one of the four conditions in which they were exposed to the different scenarios. They were asked to put themselves into the situation and to carefully read the description of the respective smart emergency detection system. Afterwards, they had to state to what extent they would accept the deployment of such a system at their workplace.

**Table 1.** Features of a privacy-invading and a privacy-preserving monitoring system.

| Feature | Privacy-Invading System | Privacy-Preserving System |
|---|---|---|
| Audio recording | Yes | Only if anomaly detected |
| Video recording | Yes | Only if anomaly detected |
| Employee identification | Yes | No |
| Data storage | Yes | No |
| Forwarding to third parties | Yes (to analyze data) | No |
| Access to the system | Security officer, manufacturer (e.g., to install updates) | Only in-house security officer |

**Table 2.** Features of a monitoring system with high and low rescue value.

| Feature | High Rescue-Value | Low Rescue-Value |
|---|---|---|
| Reaction | Immediately | After re-examination of the situation |
| Emergency call | Direct call of emergency forces (e.g., police, fire department) | Alarm signal to the in-house security officer |
| Transmission of detected information (e.g., how many people are in the building) | Yes | No |
| Appropriate instructions via loudspeaker system | Yes | No |

### 3.2. Measures

After some questions regarding the working relationship (e.g., "How long have you been employed by your current employer?") participants were asked to indicate to what extent they trust their employer. The measurements for trust [52] were adopted from the guidelines for measuring trust in organizations and contained six items (e.g., "This organization treats people like me fairly and justly."). Four additional items from Bol et al. [10] were modified from 'trust in websites' to 'trust in employer' (e.g., "My employer handles my personal information confidentially."). Trust was assessed on a seven-point Likert-scale (from 1 = I do not agree at all to 7 = I totally agree). After confirmatory factor analysis (CFA) was conducted, six items remained ($M = 5.75$, $SD = 1.1$) with Cronbach's $\alpha = 0.89$ and McDonald's $\omega = 0.9$. The Average percentage of variation explained among the items (AVE) was 0.61. Next, participants had to state to what extent they would accept a deployment of the presented system on the basis of three self-generated items on a seven-point Likert-scale ($M = 4.63$, $SD = 1.63$) with $\alpha = 0.81$, $\omega = 0.82$ and AVE = 0.62. Subsequent, perceived risks were assessed by the measurement from Bol et al. [10] on a seven-point Likert-scale (from 1 = I do not agree at all to 7 = I totally agree). The scale consisted of 10 items (e.g., "I think that the smart emergency detection system is collecting information about me.") reduced to five items after CFA ($M = 5.88$, $SD = 1.18$) with $\alpha = 0.89$, $\omega = 0.89$ and AVE = 0.61. The measurements for perceived benefits on a seven-point Likert-scale (from 1 = I do not agree at all to 7 = I totally agree) were based on the studies from Dienlin and Metzger [17] and Bol et al. [10]. The scale consisted of 10 items (e.g., "The deployment of the smart emergency detection system at my workplace serves my protection."). After CFA, five items remained ($M = 3.95$, $SD = 1.41$) with an excellent internal consistency ($\alpha = 0.93$, $\omega = 0.92$) and an AVE = 0.65. Privacy concerns were assessed on a seven-point Likert-scale (from 1 = I do not agree at all to 7 = I totally agree) via 10 items (e.g., "I'm concerned that companies are collecting too much personal information about me")

developed by Smith, Milberg and Burke [53]. CFA indicated to reduce the scale to 6 items (*M* = 5.33, *SD* = 2.09) with $\alpha$ = 0.98, $\omega$ = 0.98 and AVE = 0.87. Finally, we assessed the value of privacy. The scale from a long-term study by Trepte and Masur [47] contained 11 items such as "The protection of privacy should be enshrined in the constitution" on a seven-point Likert-scale (from 1 = I do not agree at all to 7 = I totally agree). However, after CFA with five remaining items (*M* = 4.85, *SD* = 1.28), $\alpha$ = 0.79 and $\omega$ = 0.78 we decided to exclude the scale from further calculations due to a low AVE = 0.41.

## 4. Results

All statistical analyses were computed using the statistics software SPSS Statistics 25 and SPSS Amos 25 (IBM, Armonk, New York, NY, USA). Hypotheses H1–H8 were analyzed in a path model using observed variables and maximum likelihood estimation. Indirect effects were tested using bias corrected 1000 bootstrap resamples (95% confidence interval (CI)). Bivariate correlations of the independent and dependent variables can be seen in Table 3.

### 4.1. Decriptive Values

Descriptive values of the main constructs (see Table 3) revealed that participants showed high trust in the employer (*M* = 5.75, *SD* = 1.1). Additionally, perceived risks were high (*M* = 5.88, *SD* = 1.18) while participants perceived the monitoring system as medium beneficial (*M* = 3.95, *SD* = 1.41). The means of the privacy concerns (*M* = 5.33, *SD* = 2.09) were rather high. Finally, there was a medium acceptance of the monitoring system (*M* = 4.63, *SD* = 1.63). Descriptive analyses for each condition can be found in Appendix A.

**Table 3.** Means, standard deviations and bivariate correlations of all independent and dependent variables.

| | | *M (SD)* | 1 | 2 | 3 | 4 | 5 | 6 | 7 |
|---|---|---|---|---|---|---|---|---|---|
| 1. | Trust | 5.75 (1.10) | - | | | | | | |
| 2. | Amount of tracking | - | −0.01 | - | | | | | |
| 3. | Rescue value | - | −0.01 | −0.01 | - | | | | |
| 4. | Benefits | 3.95 (1.41) | 0.14 ** | −0.24 ** | −0.04 | - | | | |
| 5. | Risks | 5.88 (1.18) | 0.00 | 0.21 ** | 0.05 | −0.40 ** | - | | |
| 6. | Privacy Concerns | 5.33 (2.09) | 0.09 * | −0.01 | 0.00 | 0.04 | 0.11 ** | - | |
| 7. | Acceptance of the system | 4.63 (1.63) | 0.17 ** | −0.40 * | 0.00 | 0.68 ** | −0.33 ** | 0.03 | - |

Note: * $p < 0.05$, ** $p < 0.01$.

### 4.2. Path Model

To evaluate the model fit, the following fit indices were used. Because in large samples, the $X^2$ test likely becomes significant [54] a ratio of $X^2$/df < 5 was chosen [55]. Hu and Bentler (1999) [56] suggested cut-off criteria of above 0.95 for Tucker-Lewis Index (TLI) and comparative fit index (CFI). The root mean square of error approximation (RMSEA) should not exceed values of 0.08 [57] and the standardized root mean squared residual (SRMR) should not be higher than 0.05 [58]. Testing the hypothesized relationships within a structural equation model (SEM) revealed a well-fitting model ($X^2$(6) = 16.64, *p* = 0.011, $X^2$/df = 2.77, CFI = 0.98, TLI = 0.93, RMSEA = 0.05 (90% CI: 0.02, 0.08), SRMR = 0.03). The whole model can be seen in Figure 2. Indirect effects were calculated using bootstrapping (*N* = 2000) with bias-corrected 95% confidence intervals [59].

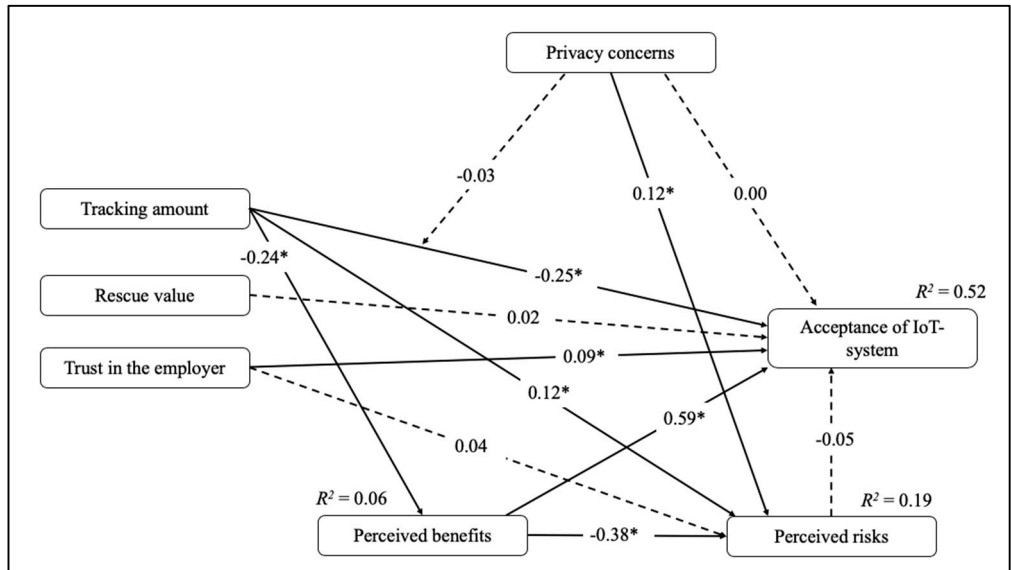

**Figure 2.** Path model with standardized effect sizes of the main model. Dashed lines indicate that the effect was not significant. The model fit was: $X^2(6) = 16.64$, $p = 0.011$, $X^2/df = 2.77$, CFI = 0.98, TLI = 0.93, RMSEA = 0.05 (90% CI: 0.02, 0.08), SRMR = 0.03. Note: * $p < 0.001$.

According to the data, trust is significantly related to the system's acceptance ($\beta = 0.09$, $p < 0.001$, $B = 0.13$, $SE = 0.04$, 95% CI [0.05, 0.22]) supporting H1. Employees who have a trusting relationship with their organization, are more willing to accept the deployment of an IoT monitoring system at the workplace which is capable of tracking their personal data. Therefore, trust is an important determinant of the acceptance of an IoT monitoring system. The relationship between trust and perceived risks is not significant ($\beta = 0.04$, $p = 0.233$, $B = 0.05$, $SE = 0.04$, 95% CI [−0.03, 0.13]) so that H2 has to be rejected. The data show that the amount of tracking is negatively related to system's acceptance ($\beta = −0.25$, $p < 0.001$, $B = −0.82$, $SE = 0.09$, 95% CI [0.65, 1.01]) supporting H3a and positively related to perceived risks ($\beta = 0.12$, $p < 0.001$, $B = 0.28$, $SE = 0.09$, 95% CI [−0.48, −0.12]) supporting H3b. Concerning the fourth hypothesis, no support is found for H4: the rescue value of the smart monitoring system and its acceptance are not significantly related to each other ($\beta = 0.02$, $p = 0.5$, $B = 0.06$, $SE = 0.09$, 95% CI [−0.11, 0.23]). Furthermore, results find support for H5a revealing that perceived benefits are positively related to system's acceptance ($\beta = 0.59$, $p < 0.001$, $B = 0.67$, $SE = 0.03$, 95% CI [0.61, 0.74]). There is no significant relationship between perceived risks and system acceptance ($\beta = −0.05$, $p = 0.107$, $B = −0.07$, $SE = 0.04$, 95% CI [−0.15, 0.02]). H6 is supported since perceived benefits are negatively related to perceived risks ($\beta = −0.38$, $p < 0.001$, $B = −0.31$, $SE = 0.03$, 95% CI [−0.38, −0.25]). There is no significant moderating effect on the relationship of tracking ability and system's acceptance ($\beta = −0.03$, $p = 0.287$, $B = −0.05$, $SE = 0.04$, 95% CI [−0.13, 0.04]); thus, H7 has to be rejected. However, the data support H8 revealing that the relationship between perceived risks and privacy concerns ($\beta = 0.12$, $p < 0.001$, $B = 0.07$, $SE = 0.02$, 95% CI [0.03, 0.11]) is significantly positive. Altogether, 52% of the variance of the system's acceptance can be explained by the tracking amount, perceived risks and benefits. All paths estimated in the model can be seen in Table 4.

In addition to the direct effects, indirect effects were tested. There are significant indirect effects of the tracking amount on perceived risks ($\beta = −0.09$, $p = 0.001$, $B = −0.22$, $SE = 0.04$, 95% CI [−0.30, −0.14]) and on perceived benefits ($\beta = 0.15$, $p = 0.001$, $B = 0.50$, $SE = 0.08$, 95% CI [0.35, 0.66]).

**Table 4.** Estimates of all paths included in the model.

| Path | B | *p* | B | SE | 95% Confidence Interval | |
|---|---|---|---|---|---|---|
| | | | | | Lower Bound | Higher Bound |
| Tracking → Benefits | −0.24 | * | −0.69 | 0.11 | 0.47 | 0.90 |
| Tracking → Risks | 0.12 | * | 0.28 | 0.09 | −0.48 | −0.12 |
| PC → Risks | 0.12 | * | 0.07 | 0.02 | 0.03 | 0.11 |
| Trust → Risks | 0.04 | 0.233 | 0.05 | 0.04 | −0.03 | 0.13 |
| Benefits → Risks | −0.38 | * | −0.31 | 0.03 | −0.38 | −0.25 |
| Risks → Acceptance | −0.05 | 0.107 | −0.07 | 0.04 | −0.15 | 0.02 |
| Benefits → Acceptance | 0.59 | * | 0.67 | 0.03 | 0.61 | 0.74 |
| Tracking → Acceptance | −0.25 | * | −0.82 | 0.09 | 0.65 | 1.01 |
| Rescue value → Acceptance | 0.02 | 0.498 | 0.06 | 0.09 | −0.11 | 0.23 |
| PC → Acceptance | 0.00 | 0.914 | 0.00 | 0.02 | −0.04 | 0.04 |
| Moderation of PC on relationship between tracking and acceptance | −0.03 | 0.287 | −0.05 | 0.04 | −0.13 | 0.04 |
| Trust → Acceptance | 0.09 | * | 0.13 | 0.04 | 0.05 | 0.22 |

Note: PC = privacy concerns, * $p < 0.001$.

## 5. Discussion

The current study examines employees' acceptance of IoT technology at the workplace that is capable of data tracking. Therefore, we test whether trust in the employer, perceived privacy risks and anticipated benefits of the IoT system are related to its acceptance by juxtaposing privacy-preserving and privacy-invading approaches. Furthermore, we investigate the moderating effect of privacy concerns. Privacy calculus [5] and CPM [6] serve as the main theoretical foundations of this research.

As suggested in the first hypothesis, trust in the employer is related to the acceptance of an IoT monitoring system deployed at the workplace. This means that employees who have a trusting relationship with their organization will more likely accept the deployment of an IoT system, even if the system is capable of collecting their personal data. Although it might seem plausible that employees pretend to accept new technology due to unbalanced power relations between management and staff the results of the path model demonstrate an explicit effect of trust on acceptance. The relationship between the level of trust in the employer and the level of IoT acceptance is highly significant. Therefore, we can assume that trust is a crucial factor of acceptance. This means that individuals who trust the employer will be more likely to accept the technology, while those who have less trust in the employer show lower acceptance—independent of the potential domination of the employer. These findings are in line with previous research stating that the willingness of individuals to provide private information is higher when they have a trusting relationship with their employer [11].

Opposed to H2, however, the results show that trust in the employer does not lead to a lower perception of potential privacy risks. This could be due to the work-related context. At the workplace employees might expect more severe consequences when personal information is collected. Moreover, individuals might differentiate between their trust in the employer and their trust in the IoT technology, meaning that they believe in a confidential handling of their data by the employer but not by the system, possibly worrying about unlimited data collection and data forwarding. Consequently, the privacy risks that employees perceive with regard to a smart monitoring system might be high, despite the existence of a trusting relationship with the employer. This, again, emphasizes the importance of trust in the employer or the organization, as employees would accept a tracking system notwithstanding its privacy threatening potential as long as they trust their employer which might explain the missing relationship between trust and perceived risks. Recent studies [8,10] found that trust in a website positively influences the willingness to provide one's personal data and therefore accept data collection. However, a substantial difference between data collection online and data tracking by an implemented IoT monitoring system is that on websites people voluntarily provide information actively indicating their data into various kinds of forms and entry fields. This visualization might contribute to the feeling of having control over collected data, while smart technology lacks transparency regarding purpose and amount of gathered information with perceived loss of control, which might lead to uncertainty

and mistrust. At this point, communication with the employees plays a decisive role. Thomas et al. [60] found that providing employees with relevant and adequate information has a positive effect on employees' trust in supervisors and management. Accordingly, a deliberate communication strategy with employees might reduce potential negative consequences on commitment and trust when implementing the tracking system.

With regard to H3, it could be confirmed that the amount of tracking has an effect on the acceptance of IoT monitoring systems and on the perception of risks. Employees tend to accept a privacy-preserving IoT system that does not store data and omits identification of individuals rather than a privacy-invading IoT system with unlimited data collection and data forwarding. These results go in line with prior findings [61] and give support to the information limit postulated by Sutanto et al. [31]. Thus, people are rather willing to provide information to an IoT system, which, by its privacy-preserving approach, does not reach their information limit. Privacy-invading technology, on the other hand, can easily exceed the information limit with a low system acceptance as a consequence. The results also reinforce the CPM [6] since privacy-preserving technology is reconcilable with the privacy management principles of this theory. In particular, this means that as long as employees' privacy is not violated by IoT technology they can maintain their privacy rules and privacy boundary turbulences do not occur. On the contrary, when employees perceive their privacy being invaded by the monitoring system they might lose the feeling of control and possession of their private information resulting in a conflict regarding information disclosure and, therefore, a lower acceptance of a privacy-invading system. Additionally, the positive relationship between the amount of tracking and the perceived risks demonstrates that people connect IoT technology that is capable of collecting their data with privacy threat supporting results from previous studies [32]. This means that even if employees cannot influence the deployment of IoT technology it is in the interest of the employer to protect the privacy of staff by implementing a privacy-preserving system in order to ensure employees' acceptance and commitment.

Furthermore, the impact of the rescue value of the IoT system is investigated. Basing on the assumptions of privacy calculus theory [5], the rescue value of the smart monitoring system represented one particular benefit of the deployment of the system since it immediately reacts when the sensors detect an emergency and directly contacts emergency forces in order to ensure the highest possible security of staff. Accordingly, H4 suggests that IoT technology with a high rescue value is related to a higher system acceptance as employees will put the perceived security provided by the system with a high rescue value over the perceived privacy threat. However, there is no relationship between the acceptance of the IoT system and its rescue value. One possible explanation is that employees already expect their workplace to provide a high level of security and do not see a particular benefit in the deployment of additional technology. Moreover, the rescue value of a smart monitoring system might be too abstract and theoretical to be perceived as a distinctive benefit of the system. Previous studies investigated more concrete or immediate benefits in the privacy calculus such as the free use of a website [35] or convenience provided by IoT devices [37]. Thus, the rescue value of a smart monitoring system might be perceived as less present compared to the privacy risks of the system, and therefore, less relevant in the risks-benefits trade-off. This assumption is substantiated by the fact that in this study benefits of the monitoring system were perceived rather low.

Regarding the impact of perceived benefits, it is shown that when people recognize the system as advantageous (e.g., in terms of a faster rescue in case of emergency) their acceptance of the IoT system is higher. A higher perception of risks, however, does not result in a lower acceptance of the system. This means that even individuals who believe the handling of the system with personal data to be problematic still accept this technology. This inference indicates that perceived benefits suppress the potential impact of perceived privacy risks. Consequently, employees are willing to accept monitoring technology as long as they evaluate these kinds of systems to be sufficiently beneficial, notwithstanding possible risks. It might be that, as employees' possibilities for actions regarding the deployment of new technology are limited at the workplace they might feel forced to acquiesce IoT monitoring systems.

In other words, if the organization decides to install the system, employees will either have to cope or quit their job. Therefore, it seems reasonable that employees would rather give their consent to collection of their data than losing their job. Additionally, as the data give evidence that individuals' perception of the system's privacy risks is high, another reason could be resignation. In this case, acceptance would be a result of a situation in which employees do not feel in control of the decision regarding technology deployment at the workplace due to unbalanced power-relations. Furthermore, legal restrictions of data tracking at the workplace might give reason to confidence, meaning that employees would accept new technology despite of the perceived privacy risks basing on their belief that their privacy is protected by jurisdiction.

Regarding H6, the data reveals that perceived benefits are negatively related to perceived risks. This means that when people perceive privacy risks as predominant they will see fewer benefits in the deployment of the system whereas when they mainly perceive the advantages of the technology they will rather perceive it to be less threatening in terms of privacy. This finding is of particular interest since it demonstrates that individuals do the risk-benefit trade-off even in situations, where their possibilities for actions are limited as it is the case at work. These results contribute to the privacy calculus research by supporting its applicability in the context of smart technology deployment. Just as people do the risk-benefit trade-off when deciding whether to disclose personal information online or not [10,17,35,36], they compare anticipated advantages and the possible privacy threat of IoT technology that is capable of tracking their data at the workplace. Thus, the privacy calculus takes place. However, in contrast to other situations in the work-related context the trade-off only leads to a behavioral intention when benefits are predominant.

In accordance with hypothesis H7, moderation effects were tested. The results did not support the assumption that privacy concerns moderated the relationship between the tracking amount and the acceptance of the IoT technology. Considering that extensive tracking of user data has become ubiquitous, people possibly perceive data collection as part of their everyday lives or the price they have to pay when using online services and smart devices. Thus, individuals might still worry about their privacy nonetheless accepting their data being tracked when their desire to use a particular device or service, such as smartphone navigation for example, exceeds the concerns regarding their privacy which in this case would be the tracking of their location [61]. Since the study was conducted in a work-related context another reason might be the resignation of employees regarding their general privacy at work. The organization not only has person-specific information of staff at its disposal, but also the decisional power regarding the deployment of monitoring technology capable of data tracking. This means that the only two options left for the employees are to either tolerate the data collection notwithstanding their privacy concerns or to quit their job in order to evade being exposed to the frequent monitoring. This corroborates the findings of Wirth et al. [62] who showed that resignation has a positive effect on the perception of benefits and a negative effect on the perception of risks. Consequently, when employees react to privacy threats with resignation, they might have an altered perception of risks and benefits explaining a higher acceptance of monitoring technology despite of possible privacy concerns. Moreover, it should be noted that the items from the privacy concerns scale were adapted to the workplace context and measured privacy concerns regarding the handling of personal data by the employer and not by the IoT system. With the CPM principles in mind, it is conceivable that individuals are able to apply privacy rules at work but at the same time experience privacy boundary turbulences caused by the deployment of the monitoring system meaning that employees are less worried about their privacy at work than about a third party in form of an external system getting access to their data. In this case, privacy concerns might moderate the relationship between the amount of tracking and the acceptance of the monitoring system when the privacy concerns scale refers to the system instead of the employer. However, there is an indirect effect of privacy concerns on system acceptance. People who are worried about their privacy might also have a higher perception of privacy risks, which according to the results are directly related to the system's acceptance explaining the indirect effect.

Interestingly, privacy concerns are positively related to perceived risks (H9). Individuals who are worried about their privacy are also more sensitive regarding their perception of potential risks for their personal data. Consequently, implications which can be drawn for employers are: first, that it is in their interest to protect privacy of staff by limiting employee monitoring or deploying technology with the privacy-by-design approach. In other words, considering the privacy of the workforce before implementing an IoT system enables the employer to resort to a system already working in a privacy-preserving way by its technical implementation which, thus, is more likely to be accepted by the employees. This decision might be a confidence-building measure ensuring a responsible handling of employees' data respecting their value of privacy. Second, if the decision is in favor of installing a system that does not automatically cover the privacy of the employees by virtue of its functioning, and thus, does not have a privacy-by-design approach, the employer nevertheless has certain possibilities of influencing the acceptance of the system. On the one hand, the employer could provide the employees with detailed information regarding purpose and reasons for collecting the data. On the other hand, the employer could emphasize the benefits, such as security, that the system brings to the employees. Any measures that help to increase the acceptance of the new technology in the company are fundamentally beneficial in order not to jeopardize trust, commitment and performance.

*Limitations and Future Research*

Some limitations of the study must be noted. First, participants did not evaluate a real IoT system, but had to imagine a hypothetical situation where such a system is implemented at their workplace by reading descriptions of the monitoring technology. Such scenarios are functional by allowing to draw first conclusions as well as making comparisons between different conditions. However, it is important to note that the generalizability is reduced due to the artificial content of the presented vignettes. In order for the evaluation of these scenarios to be realistic, the vignettes of this study describe existing features of smart monitoring systems and provide employees with information they can access due to data protecting legislation, such as the GDPR. However, employees might react differently if their employer actually makes the decision to deploy such a system. For this reason, in further studies research needs to reflect on employees' reactions and acceptance processes under real circumstances. Furthermore, concerning the different scenarios caution should be taken when interpreting the results as the descriptions were written based on existing system features; however, in order to create different conditions the features were isolated. Thus, the interaction of the tracking amount and the rescue value are only hypothetical. Caution should also be taken with regard to generalizability of IoT system acceptance. Due to the heterogeneity and various characteristics of IoT devices, IoT systems might be evaluated differently in terms of privacy, perceived risks and benefits. Another remark is the unequal gender distribution of women and men of about 70:30, which is worth considering in future studies. Furthermore, the online study could not measure real behavior, but only estimate to which degree participants would accept the technology. Therefore, future studies could measure physiological effects or changes in work performance of employees when being monitored at work. A methodological limitation, as mentioned in the discussion, is that privacy concerns were measured only with regard to the employer and not to the IoT system. In order to better understand employees' evaluation of smart technology it would be beneficial to also include this measurement into research. Since this study examined IoT technology in a work-related context it would also be interesting to investigate how people interact with the same technology in their private homes compared to the workplace. When thinking about privacy, the awareness of data tracking might be also examined as a determinant of the usage of IoT. As results by Thomas et al. [60] demonstrated that providing employees with information influences their trusting relationship to management and supervisors, awareness and knowledge of employees regarding IoT monitoring systems at the workplace should therefore be included in future investigations.

## 6. Conclusions

The current study aimed to re-examine the privacy calculus in order to investigate whether employees do the risk-benefit trade-off even when their possibilities for actions are limited, as it is the case at the workplace. Furthermore, we tested the applicability of privacy calculus within the framework of IoT technology usage. Moreover, we investigated the necessary preconditions for employees' acceptance of the deployment of such a technology that captures their data, additionally embedding the results in the CMP theory.

In this paper, we employed a $2 \times 2$ between-subjects online experiment ($N = 661$) and examined employees' acceptance of a smart emergency detection system, depending on whether the system's tracking is privacy-invading or privacy-preserving as well as on the rescue value of the system. While the amount of tracking and trust in the employer had a significant influence on the acceptance, no effect of the system's rescue value or perceived risks could be found. However, perceived benefits could be confirmed as a predictor of acceptance. Privacy concerns did not affect system acceptance but were related to perceived risks.

This study contributes to privacy literature by exploring employees' privacy at the workplace considering possible vulnerabilities due to the deployment of IoT technology capable of data tracking. The results support the application of privacy calculus as a theoretical foundation when using smart technology. Findings of this research are supposed to reflect on the role of innovative applications of IoT technology at the workplace. Furthermore, the results and implications discussed in our study are meant to contribute to the identification of the determinants that affect smart technology acceptance. In particular, the amount of tracked data as well as trust in the employer and perceived benefits of the IoT monitoring system are significant factors regarding the acceptance of smart technology. The resulting implications for employers are to communicate changes regarding technological development at the workplace and to implement privacy-by-design when deploying smart technology capable of data tracking in order to ensure acceptance of this technology.

**Author Contributions:** This paper describes research undertaken by E.P. in the framework of the PhD studies, under the supervision of N.C.K. at the University of Duisburg-Essen. Data collection, analysis, and formulation of research paper was undertaken by E.P., N.C.K. supervised research methodology and reviewed paper formulation.

**Funding:** This research is funded by Evonik Industries, research project "Privacy mechanisms for collection and analysis of person-related data".

**Conflicts of Interest:** The authors declare no conflict of interest.

## Appendix A. Scenarios and Descriptive Analyses for All Conditions

*Appendix A.1. Privacy-Preserving, High Rescue Value*

The intelligent monitoring system is designed to recognize accidents and emergencies at work and limit consequential damage by, among others, alerting auxiliaries faster. The objective is to collect as little personal data as possible. For this reason, the system initially works in privacy mode, in which neither audio nor video recordings are made. Instead, sensors analyze ground vibrations to detect movements and the air quality so that gas or fire, for example, can be detected immediately. Sound and video are only recorded when the system detects deviations (e.g., when a person falls down, people run or the air composition changes). The system determines within a very short time whether an emergency has occurred without identifying the employees. The recorded data is not stored by the system and is not passed on to third parties. Only the company's internal security officer has access to the system.

If the system has been able to detect an emergency, it reacts immediately by an emergency call that is made directly to the emergency services. The video evaluation allows the system to determine what kind of emergency happened. In the event of fire, the fire brigade will be informed, the police in the event of a burglary. In addition to the address, the system provides all relevant information regarding the emergency to the auxiliaries. Therefore, auxiliaries are able to be best prepared for the

rescue mission. At the same time, the system sends instructions (e.g., how employees should behave) according to the emergency via loudspeakers and escorts people out of the building.

162 participants aged between 18 and 59 ($M = 28.06$, $SD = 7.55$) were assigned to this condition. The sample included 106 females, 53 males and three individuals who did not specify their gender. Descriptive values for Condition 3 (privacy-preserving, high rescue value) can be seen in Table A1.

**Table A1.** Means, standard deviations and bivariate correlations of all independent and dependent variables for Condition 3.

|  | *M (SD)* | 1 | 2 | 3 | 4 | 5 |
|---|---|---|---|---|---|---|
| 1. Trust | 5.38 (1.11) | - | | | | |
| 2. Benefits | 4.83 (1.18) | 0.14 | - | | | |
| 3. Risks | 5.0 (1.13) | −0.03 | −0.44 ** | - | | |
| 4. Privacy Concerns | 4.93 (1.38) | 0.12 | 0.03 | 0.21 ** | - | |
| 5. Acceptance of the system | 5.27 (1.48) | 0.18 * | 0.67 ** | −0.38 ** | −0.06 | −0.05 |

Note: * $p < 0.05$, ** $p < 0.01$.

### *Appendix A.2. Privacy-Preserving, Low Rescue Value*

The intelligent monitoring system is designed to recognize accidents and emergencies at work and limit consequential damage by, among others, alerting auxiliaries faster. The objective is to collect as little personal data as possible. For this reason, the system initially works in privacy mode, in which neither audio nor video recordings are made. Instead, sensors analyze ground vibrations to detect movements and the air quality so that gas or fire, for example, can be detected immediately. Sound and video are only recorded when the system detects deviations (e.g., when a person falls down, people run or the air composition changes). The system determines within a very short time whether an emergency has occurred without identifying the employees. The recorded data is not stored by the system and is not passed on to third parties. Only the company's internal security officer has access to the system.

If the system was able to detect an emergency, it will first re-examine the circumstances in order to explicitly declare the situation as an emergency. After that, the system sends a signal to the company's security service. A red light alerts the security personnel signaling that help is needed. Based on this situation, the security personnel can intervene themselves or inform the fire brigade and police emergency service.

164 participants aged between 18 and 55 ($M = 27.16$, $SD = 7.03$) were assigned to this condition. The sample included 110 females, 52 males and two individuals who did not specify their gender. Table A2 includes descriptive values for Condition 4 (privacy-preserving, low rescue value).

**Table A2.** Means, standard deviations and bivariate correlations of all independent and dependent variables for Condition 4.

|  | *M (SD)* | 1 | 2 | 3 | 4 | 5 |
|---|---|---|---|---|---|---|
| 1. Trust | 5.42 (1.01) | - | | | | |
| 2. Benefits | 4.69 (1.17) | 0.16 * | - | | | |
| 3. Risks | 5 (1.1) | 0.03 | −0.35 ** | - | | |
| 4. Privacy Concerns | 4.81 (1.49) | 0.03 | 0.00 | 0.16 * | - | |
| 5. Acceptance of the system | 5.33 (1.3) | 0.04 | 0.65 ** | −0.35 ** | −0.06 | −0.2 ** |

Note: * $p < 0.05$, ** $p < 0.01$.

### *Appendix A.3. Privacy-Invading, High Rescue Value*

The intelligent monitoring system is designed to recognize accidents and emergencies at work and limit consequential damage by, among others, alerting auxiliaries faster. With the help of sensors as well as audio and video surveillance, the system makes real-time recordings, using algorithms that continuously evaluate whether the normal case exists (everything happens as usual) or an emergency

has occurred (e.g., if a person falls down or sensors detect a gas rise in the air). Through the help of camera sensors, the system is able to detect the employees in order to provide as detailed information as possible to auxiliaries. The recorded data is stored for later evaluation. This data can be passed to third parties, for example, for training or analysis purpose. The system access is granted to the security officer of the company as well as the manufacturer to be able to, among other things, install system updates.

If the system has been able to detect an emergency, it reacts immediately by an emergency call that is made directly to the emergency services. The video evaluation allows the system to determine what kind of emergency happened. In the event of fire, the fire brigade will be informed; the police in the event of a burglary. In addition to the address, the system provides all relevant information regarding the emergency to the auxiliaries. Therefore, auxiliaries are able to be best prepared for the rescue mission. At the same time, the system sends instructions (e.g., how employees should behave) according to the emergency via loudspeakers and escorts people out of the building.

170 participants aged between 18 and 65 ($M = 27.79$, $SD = 7.38$) were assigned to this condition. The sample included 120 females, 47 males and three individuals who did not specify their gender. Below (Table A3), descriptive values for Condition 1 (privacy-invading, high rescue value) are shown.

**Table A3.** Means, standard deviations and bivariate correlations of all independent and dependent variables for Condition 1.

| | M (SD) | 1 | 2 | 3 | 4 | 5 |
|---|---|---|---|---|---|---|
| 1. Trust | 5.42 (1.15) | - | | | | |
| 2. Benefits | 4.31 (1.3) | 0.14 | - | | | |
| 3. Risks | 5.65 (1.12) | −0.13 | −0.43 ** | - | | |
| 4. Privacy Concerns | 4.84 (1.6) | −0.18 * | 0.06 | 0.23 ** | - | |
| 5. Acceptance of the system | 4.04 (1.6) | 0.18 * | 0.68 ** | −0.48 ** | 0.01 | −0.26 ** |

Note: * $p < 0.05$, ** $p < 0.01$.

*Appendix A.4. Privacy-Invading, Low Rescue Value*

The intelligent monitoring system is designed to recognize accidents and emergencies at work and limit consequential damage by, among others, alerting auxiliaries faster. With the help of sensors as well as audio and video surveillance, the system makes real-time recordings, using algorithms that continuously evaluate whether the normal case exists (everything happens as usual) or an emergency has occurred (e.g., if a person falls down or sensors detect a gas rise in the air). Through the help of camera sensors, the system is able to detect the employees in order to provide as detailed information as possible to auxiliaries. The recorded data is stored for later evaluation. This data can be passed to third parties, for example, for training or analysis purpose. The system access is granted to the security officer of the company as well as the manufacturer to be able to, among other things, install system updates.

If the system was able to detect an emergency, it will first re-examine the circumstances in order to explicitly declare the situation as an emergency. After that, the system sends a signal to the company's security service. A red light alerts the security personnel signaling that help is needed. Based on this situation, the security personnel can intervene themselves or inform the fire brigade and police emergency service.

165 participants aged between 18 and 65 ($M = 27.68$, $SD = 8.07$) were assigned to this condition. The sample included 111 females and 54 males. Table A4 contains descriptive values for Condition 2 (privacy-invading, low rescue value).

**Table A4.** Means, standard deviations and bivariate correlations of all independent and dependent variables for Condition 2.

|  | *M (SD)* | 1 | 2 | 3 | 4 | 5 |
|---|---|---|---|---|---|---|
| 1.  Trust | 5.40 (1.1) | - | | | | |
| 2.  Benefits | 4.2 (1.2) | 0.19 * | - | | | |
| 3.  Risks | 5.71 (0.96) | −0.17 * | 0.23 ** | - | | |
| 4.  Privacy Concerns | 4.92 (1.54) | 0.07 | 0.01 | 0.12 | - | |
| 5.  Acceptance of the system | 3.93 (1.58) | 0.22 ** | 0.7 ** | −0.32 ** | 0.07 | −0.33 ** |

Note: * $p < 0.05$, ** $p < 0.01$.

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
