# Peer review of "Acceptance of Smart Electronic Monitoring at Work as a Result of a Privacy Calculus Decision"

_informatics, doi:10.3390/informatics6030040_

Round 1

Reviewer 1 Report

Technical Aspects

The paper technically does a great job explaining the theory, setting up the hypotheses and experimental design, creating a randomized sample, and inserting measures backed up by prior research along with reliability analyses.  The sample size is appropriate for the type of data analysis completed but the male-dominated sample was problematic and uncorrectable for the purposes of this potential publication.

Design Aspects

The paper acknowledged that this is not a real life analysis but rather a simulation.  Is the simulation representative of what actually might happen in an organization? The scenarios (well written with consistent wording) have a high level of detail about smart electronic monitoring systems.  The scenarios are directly associated with the variables in the research study.  The scenarios might be a lot of reading for an online case that could lead to rushed and inappropriate answers but the length is necessary due to the research content required. 

Simulations, by design, are limited in what they can cover.  Viewing the simulations from the perspective of the employee, I have difficulty with understanding whether employees would know details about the monitoring systems and would care about those details. The basic simulation starts by asking participants to put themselves into the situation and understand the emergency detection system. You should check the research to see if employees really learn that much about the systems?  Do they get this information in privacy policies, employee handbooks, Greybooks, management e-mails, meetings, or nowhere?  The results of your checks could be included in the future research section.

Theories such as privacy calculus, communication privacy management, and the technology acceptance model have limited discussion on how management communicates with employees about the nature of privacy with regard to monitoring.  I recommend that in the "future research" section on page 10, line 42, the nature and amount of communication employers have concerning the privacy associated with the smart technologies might have an impact on at least trust with management.  A useful source on this topic is the following.

The Central Role of Communication in Developing Trust and Its Effect on Employee Involvement

Gail Fann Thomas, Roxanne Zolin, Jackie L. Hartman

First Published March 27, 2009 Research Article

https://doi.org/10.1177/0021943609333522

Communication plays an important role in the development of trust within an organization. While a number of researchers have studied the relationship of trust and communication, little is known about the specific linkages among quality of information, quantity of information, openness, trust, and outcomes such as employee involvement. This study tests these relationships using communication audit data from 218 employees in the oil industry. Using mediation analysis and structural equation modeling, we found that quality of information predicted trust of one's coworkers and supervisors while adequacy of information predicted one's trust of top management. Trust of coworkers, supervisors, and top management influenced perceptions of organizational openness, which in turn influenced employees' ratings of their own level of involvement in the organization's goals. This study suggests that the relationship between communication and trust is complex, and that simple strategies focusing on either quality or quantity of information may be ineffective for dealing with all members in an organization.

Other missing considerations that might be mentioned include the voluntariness and social influence associated with the smart technologies (Unified Theory of Acceptance and Use of Technology), perceived ease of use (Technology Acceptance Model), and system, information, and service quality (Delone and McLean IS Success Model).  How much do employees typically know about smart systems in companies?  This would provide more support to indicating how "real life" the simulations are.

The conclusions provide a short summary of the importance of the amount of tracked data, perceived risks, and perceived benefits of the IoT system regarding acceptance of smart technology.  The paper comes short in discussing the implications for employers at the end.  If an employee would read this paper, how would he/she understand privacy-by-design (page 11, line 33)?  What is the main lesson learned from this paper from the practical point of view?  I feel there has to be a separate paragraph on the implications for employers no matter how slim they may be.

Author Response

Dear Editor, dear Reviewers,

Before we give a detailed response to the reviewers’ comments and report on the changes we have made to the manuscript, we want to thank you for the helpful suggestions and comments on our paper. Please let us emphasize that we really appreciate the high level of constructiveness that characterized all your suggestions. We tried to take up all feedback and feel that the manuscript indeed improved considerably. Most importantly, with the new version of the manuscript, we tried to offer a more fine-grained discussion of our findings and their limitations as well as we also tried to be more explicit about how our study contributes to the present state of knowledge. Based on the feedback of the reviewers we recalculated the scales and the path model resulting in a better model fit. The results can now be interpreted as an even better indication for the weighing of risks and benefits in the privacy calculus model. Below you find an outline of the revisions we have made to the manuscript in accordance with your comments.

Reviewer 1: 

The sample size is appropriate for the type of data analysis completed but the male-dominated sample was problematic and uncorrectable for the purposes of this potential publication.

Authors’ response:

Thank you for the helpful remark. We agree that a balanced sample increases the representativeness of the study and therefore acknowledged the gender ratio as a limitation of our study. Unfortunately, the imbalance between the number of men and women participating in psychological online studies is not unusual (e.g., Meier & Neubaum, 2019). Gender differences regarding privacy and the acceptance of smart technology were not to be expected (Krasnova, Veltri & Günther, 2012; Malhotra, Kim, & Agarwal, 2004). However, we conducted an independent-samples t-test to compare gender-related system acceptance.There was no significant difference in men (= 4.68, SD = 1.56) and women (= 4.62, SD = 1.67); t(651) = -.47, p= 0.636.

Reviewer 1: 

The paper acknowledged that this is not a real life analysis but rather a simulation. Is the simulation representative of what actually might happen in an organization?

Authors’ response:

You have raised an important question. As a matter of fact, intelligent tracking systems (similar to the system described in our scenarios) are widely spread in organizations but also in public spaces, such as train stations and airports. At the workplace, so called “people analytics methods seek to capture masses of quantitative data in order to reveal hidden patterns that are correlated with employee success or failure. Sometimes that data will relate solely to the employee’s job performance - namely what the employee has specifically done while acting within the scope of employment. However, it may also relate to aspects of the employee as an individual, such as the employee’s overall aptitude in various skills and settings, her health, her psychological disposition, or even what she had for breakfast.“ (Bodie, Cherry, McCormick, & Tang, 2017, p. 25). 

In order to present realistic versions of this technology, for the scenarios of our study we collected information from manufacturer websites (e.g., www.rhebo.com) as well as from existing engineering research (Virgona, Kirchner, & Alempijevic, 2015). To further enhance the representativeness of the results, only employees were addressed as this group is best able to put themselves into the respective situation.

Reviewer 1:

The scenarios might be a lot of reading for an online case that could lead to rushed and inappropriate answers but the length is necessary due to the research content required.

Authors’ response:

We appreciate that you understand the necessity of the length of the scenarios. We tried to keep them as short as possible while providing all relevant information. Further, we chose a between-person design for our study so that every participant reads only one longer text in order to avoid rushed and inappropriate answers. Additionally, before conducting our study, we tested how long reading times of participants should be (minimum 220 seconds) and excluded those with reading times under 220 seconds from our sample.

Reviewer 1:

Simulations, by design, are limited in what they can cover. Viewing the simulations from the perspective of the employee, I have difficulty with understanding whether employees would know details about the monitoring systems and would care about those details. The basic simulation starts by asking participants to put themselves into the situation and understand the emergency detection system. You should check the research to see if employees really learn that much about the systems?  Do they get this information in privacy policies, employee handbooks, Greybooks, management e-mails, meetings, or nowhere?  The results of your checks could be included in the future research section.

Authors’ response:

You are absolutely right. The scenarios only cover the ideal case, meaning that employees are provided detailed information on the implementation and the purpose of the tracking system. However, scenarios are important to understand the effect of tracking on the acceptance of smart monitoring technology and to examine whether it is perceived as beneficial or risky regarding employees’ privacy. In the real world, the degree to which information is provided to employees depends on the organization and on jurisdiction. In Europe, employers are obliged by the General Data Protection Regulation (GDPR) to indicate every collection of data. Current literature demonstrates that there are major differences regarding legal requirements throughout different countries (Atkinson, 2018) for example referring to what kind of communication (e.g., emails) or which areas in the organization the employer is allowed to monitor. Further, requirements justifying employee surveillance as well as the obligation to inform the stuff about being monitored differ. 

Following your suggestion, we added further information in to the manuscript, on the one hand discussing the impact of legal requirements on the implementation of a monitoring system at the workplace (see p. 3):

In order to raise an organization’s efficiency, workplace monitoring has become a convenient instrument for employers, promising positive outcomes such as a higher security of employees (e.g., by identifying potential safety hazards) or support of HR decision-making by providing insights about job performance. However, the vast collection of personal data raises enormous challenges for employee privacy. Intelligent IoT tracking systems are able to gather masses of sensitive data correlating it to employee performance and automatically drawing conclusions on improvable aspects regarding efficiency, which is called people analytics (Moore, Upchurch, & Whittaker, 2018). From the legal point of view, employees give up a lot of privacy expectations, since “The employer is allowed to monitor employees through supervisors, video cameras, computer software, or other methods that capture employees working within the scope of employment.” (Bodie, Cherry, McCormick, & Tang, 2017, p. 26). However, there are limits to employee surveillance, for example if monitoring is undisclosed. Secret tracking requires severe and legitimate reasons, such as suspicion of fraud or theft (Bodie et al., 2017). Generally, employees should be provided with information about being subject of data collection (e.g., by information signs). Accordingly, the commencement of the European General Data Protection Regulation (GDPR) guarantees individuals the right to be informed about the processing purposes, the legal basis or automated decision-making processes. It should be noted, however, that legal regulations between countries have to be differentiated.

We also extended the according passage in the limitation section (see p. 13):

Such scenarios are functional by allowing to draw first conclusions as well as making comparisons between different conditions. In order for the evaluation of these scenarios to be realistic, the vignettes of this study describe existing features of smart monitoring systems and provide employees with information they can access due to data protecting legislation, such as the GDPR. However, employees might react differently if their employer actually makes the decision to deploy such a system. For this reason, in further studies, research needs to reflect on employees’ reactions and acceptance processes under real circumstances. 

Reviewer 1:

I recommend that in the "future research" section on page 10, line 42, the nature and amount of communication employers have concerning the privacy associated with the smart technologies might have an impact on at least trust with management.  A useful source on this topic is the following.

The Central Role of Communication in Developing Trust and Its Effect on Employee Involvement (Thomas, Zolin, & Hartman, 2009)

Authors’ response:

We are very thankful for your literature suggestion. We have integrated the proposed paper in the discussion part. (see p. 10)

Recent studies [8, 10] found that trust in the website positively influences the willingness to provide one’s personal data and therefore accept data collection. However, a substantial difference between data collection online and data tracking by an implemented IoT monitoring system is that on websites, people voluntarily provide information actively indicating their data into various kinds of forms and entry fields. This visualization might contribute to the feeling of being in control over collected data while smart technology lacks transparency regarding purpose and amount of gathered information with perceived loss of control, uncertainty and mistrust as possible consequences. At this point, communication with the employees plays a decisive role. In their study, Thomas, Zolin and Hartman [58] found that providing employees with relevant and adequate information has a positive effect on employees’ trust in supervisors and management. Accordingly, a deliberate communication strategy with employees might reduce potential negative consequences on commitment and trust when implementing the tracking system. Therefore, when privacy-invading IoT-technology is deployed especially people with a high value of privacy might react with a decreasing commitment and experience negative emotions such as stress, considering the few options they have to evade the tracking. 

Reviewer 1:

Other missing considerations that might be mentioned include the voluntariness and social influence associated with the smart technologies (Unified Theory of Acceptance and Use of Technology), perceived ease of use (Technology Acceptance Model), and system, information, and service quality (Delone and McLean IS Success Model).  

Authors’ response:

We thank the reviewer for the helpful remark and agree that considerations regarding technology acceptance models such as TAM (Davis, 1989) and UTAUT (Venkatesh, Morris, Davis & Davis, 2003) should be included to the manuscript. We therefore added the following section into literature review (see p. 3):

With regard to technology acceptance, many theories have been proposed by scholars. The technology acceptance model (TAM; Davis, 1989) found substantial and empirical support and is well-studied in this field of research. Particularly, it aims at predicting a person’s behavioral intention to use a system as well as actual usage depending on perceived usefulness (i.e. to what extent the system helps enhancing individual’s job performance) and perceived ease of use (i.e. the required effort of the usage). A theoretical advancement of TAM and other constructs is the unified theory of acceptance and use (UTAUT, Venkatesh, Morris, Davis, & Davis, 2003). This extensive model takes additional factors, such as social influence and voluntariness of use as determinants of usage intention and behavior into consideration. At this point, it becomes clear that theories and models of technology acceptance have a common goal, which is to explain behavioral intention and usage adoption. However, traditional and advanced approaches of technology acceptance do not seem to be entirely applicable in the context under investigation here: Employees are affected by the system but apart from being monitored, they do not interact with it or actively use it. Moreover, they do not have the choice to adopt or reject the system but can only evaluate to what degree they approve the authority decision regarding the system’s implementation at the workplace. Accordingly, most constructs taken up by technology acceptance models, such as investigating the effect of system usage on job performance, are not applicable in this context. Further, in his diffusion of innovations theory Rogers (2003) describes innovation diffusion as an adaptation process where individuals first learn about the innovation’s existence, form an attitude towards it and engage with it before finally putting it into use. However, he points out, that specifically in the context of organizations, an “authority innovation-decision is one with which the organization’s employees must comply” (p. 403). In other words, in the case of the implementation of an IoT-monitoring system at the workplace, employees have no decisional power, which is why common technology acceptance models seem less suitable than the risk-benefit trade-off postulated by the privacy calculus. Last but not least, these models do not consider the privacy aspect, which, given the potentially privacy intrusive character of tracking systems, plays a decisive role for the acceptance of such systems. In the light of the above, privacy calculus serves as the main theoretical foundation investigating system acceptance as an outcome of a risk-benefit trade-off. 

Reviewer 1:

How much do employees typically know about smart systems in companies? This would provide more support to indicating how "real life" the simulations are.

Authors’ answer:

You are absolutely right, that determining how much employees know about implemented technology can enhance the simulations. In order to find out, how much information employees have about IoT systems in organizations, participants of the study must be people working in companies, which already deploy such systems. Therefore we extended the future research section with the suggestion to investigate knowledge and awareness of employees in future studies (see p. 13):

When thinking about privacy, the awareness of data tracking might be also examined as a determinant of the usage of IoT. As results by Thomas, Zolin and Hartman [58] demonstrate that providing employees with information influences their trusting relationship to management and supervisors, awareness and knowledge of employees regarding IoT monitoring systems at the workplace should be therefore included in future investigations.

Reviewer 1:

The paper comes short in discussing the implications for employers at the end. If an employee would read this paper, how would he/she understand privacy-by-design (page 11, line 33)? What is the main lesson learned from this paper from the practical point of view?  I feel there has to be a separate paragraph on the implications for employers no matter how slim they may be.

Authors’ response:

We are grateful for this suggestion as it helps adding a practical view to the study. Therefore, as proposed by the reviewer, we edited an additional paragraph with implications for employers (see p. 12):

Implications, which can be drawn for employers, are: First that it is in their interest to protect privacy of staff by limiting employee monitoring or deploying technology with the privacy-by-design approach. In other words, considering the privacy of the workforce before implementing an IoT system enables the employer to resort to a system already working in a privacy-preserving way by its technical implementation, which thus, is more likely to be accepted by the employees. This decision might be a confidence-building measure insuring a responsible handling of employees’ data respecting their value of privacy. Second, if the decision is in favor of installing a system that does not automatically cover the privacy of the employees by virtue of its functioning and thus does not have a privacy-by-design approach, the employer nevertheless has certain possibilities of influencing the acceptance of the system. On the one hand, the employer could provide his employees with detailed information regarding purpose and reasons for collecting the data. On the other hand, the employer could emphasize the benefits, such as security, that the system brings to the employees. Any measures that help to increase the acceptance of the new technology in the company are fundamentally beneficial in order not to jeopardize trust, commitment and performance.

Thank you again for the numerous insightful comments and suggestions!

Best wishes,

The authors

References

Atkinson, J. (2018). Workplace Monitoring and the Right to Private Life at Work. The Modern Law Review81(4), 688-700.

Bodie, M. T., Cherry, M. A., McCormick, M. L., & Tang, J. (2017). The Law and policy of people analytics. U. Colo. L. Rev.88, 961.

Krasnova, H., Veltri, N. F., & Günther, O. (2012). Self-disclosure and privacy calculus on social networking sites: The role of culture. Business & Information Systems Engineering4(3), 127-135.

Malhotra, N. K., Kim, S. S., & Agarwal, J. (2004). Internet users' information privacy concerns (IUIPC): The construct, the scale, and a causal model. Information systems research15(4), 336-355.

Meier, Y., & Neubaum, G. (2019). Gratifying Ambiguity: Psychological Processes Leading to Enjoyment and Appreciation of TV Series with Morally Ambiguous Characters. Mass Communication and Society. DOI: 10.1080/15205436.2019.1614195

Moore, P. V., Upchurch, M., & Whittaker, X. (2018). Humans and machines at work: monitoring, surveillance and automation in contemporary capitalism. In Humans and Machines at Work (pp. 1-16). Palgrave Macmillan, Cham.

Thomas, G. F., Zolin, R., & Hartman, J. L. (2009). The central role of communication in developing trust and its effect on employee involvement. The Journal of Business Communication (1973)46(3), 287-310.

Virgona, A., Kirchner, N., & Alempijevic, A. (2015, November). Sensing and perception technology to enable real time monitoring of passenger movement behaviours through congested rail stations. In Australasian Transport Research Forum. ATRF.

Reviewer 2 Report

The manuscript submitted to Informatics investigates potential predictors of the acceptance of different types of smart technologies in the work context. More specifically, the authors try to predict the acceptance of four types of intelligent monitoring systems designed to identify accidents or emergencies at work and limit damage. The authors implemented a 2 (privacy-invasive vs. privacy-preserving) x 2 (high rescue value vs. low rescue value) between-person design to study the influences of the systems’ characteristics on people’s acceptances of such a system. The study was designed as an online survey of 661 participants. 

Overall, the paper focuses on a relevant topic. In the light of more and more intangible privacy risks stemming from online services providers or product manufacturers, it is indeed important to understand how people perceive certain privacy-invasive aspects of new technology and what factors hinder or facilitate acceptance of such a product. I do appreciate that the authors aimed at recruiting a large sample. Given that much prior work is based on under-powered studies, it is refreshing to read a study that was able to recruit a comparatively large sample. In general, I thus would deem the paper suitable for the audience of Informatics as well as the broader community of privacy and technology scholars. 

Despite an overall good first impression, the paper has several theoretical, and empirical flaws and weaknesses. Although many of them could potentially be addressed in a thorough revision, I would nonetheless recommend a “revise & resubmit” as some of the problems require a complete reworking of the theoretical approach that would lead to a different assumptions and a different model. Before I will provide detailed comments and remarks about these issues, I also would like to note that I think that trying to understand these processes (technology/application/SNS acceptance/adoption processes) with vignettes and online surveys is of limited value. The major limitations - the hypothetical scenarios and the hypothetical acceptance - that were also acknowledged by the authors are problematic: We need to study these processes under real circumstances where we - as scholars - can observe people’s use of technology and reactions to certain manipulations. Prior studies have shown that intentions to not necessarily align with behaviors and confrontation with actual technology leads to different reactions that hypothetical scenarios. 

1. First, although the authors study technology acceptance, there is no reference to any of the established and well-studied theories that actually aim at predicting technology adoption/acceptance (e.g., TAM by Venkatesh et al.; diffusion theory by Rogers, or even the mobile phone appropriation model by Wirth et al.). This is unfortunate as these theories  could have provided more insights into actual acceptance processes and probably would even have suggested other potential predictors/antecedents of technology acceptance. This is a major flaw in the theoretical conceptualization of this study. The authors should at least discuss these approaches and situate there research accordingly, even if this would mean addressing potential neglects in the limitation section. 

2. The formulation of almost all hypotheses is imprecise and misleading given the design of the study: All hypotheses that refer to relationships between variables that could be interpreted as causal effects (based on the experimental design; e.g., H3a and H3b) are formulated “relationships". Hypotheses for relationships that cannot be interpreted causally based on the design, however, are formulated as actually effects (e.g., H1, H5). Furthermore, H5b and H6 are exactly the same. 

3. Moderation hypotheses that do not specify a direction cannot be tested. Please reformulate to include the direction of the moderating effect.

4. Despite a long literature review and a comprehensive theoretical rationale for all the hypotheses, I am not fully convinced by the proposed model and its relationships. 

First, shouldn’t the rescue value manipulation directly affect the perceived benefits. The measure has even a high compatibility with the manipulation. Why did you not include this path in the model?

Second, why should perceived benefits affect perceived risks? This does not make sense theoretically. They could be correlated, but I don’t think it makes sense to assume a causal relationship here. The indirect effect is hence not logical either. 

I am not convinced by the authors argumentation that privacy concerns should moderate the relationship between amount of tracking and technology acceptance. Did you check whether the amount of tracking affected privacy concerns? Could it be that they are similarly affect as perceived privacy risks? A lot of studies investigating the privacy calculus even assume that risks and concerns are somewhat similar. 

Please not that relationships “after” the mediators or between non-manipulated variables cannot be interpreted causally. Strictly speaking even relationships between manipulations and outcome variables can no longer be interpreted causally if that effect is controlled for other cross-sectional relationships. 

5. In general, I strongly believe in the value of openness and transparency. In light of recent meta-scientific discoveries (e.g., replication crisis, questionable research practices, etc.), I suggest to the authors to evaluate whether they have met the standards of the Peer Reviewers’ Openness Initiative (https://opennessinitiative.org/). I also strongly believe that the public availability of all materials, data, and analysis scripts is important not only for evaluating the study’s contribution and methodological soundness, but also to allow other researchers to reproduce this study's results, replicate it independently, or use it in meta-analyses. As I have no insight into the data, the actual analysis script and cannot reproduce the results based on the data at this moment, I do not feel able to evaluate the methodical soundness of this paper. 

6. Please justify your sample size. Did you compute a priori power analyses? If not, please provide results from a sensitivity analysis. 

7. Please provide information about the scales factorial validity and reliability. Please conduct confirmatory factor analyses and also provide more reasonable reliability estimates than Cronbach’s alpha (e.g., measures of composite reliability such as McDonald’s omega). 

8. It seems that the authors do not report all paths estimated in the model. The current model has only 3 degrees of freedom yet, figure 2 suggest that many more relationships were estimated. Please be comprehensive and report all estimated relationships (including potential correlation between exogenous variables). 

9. Interpreting the model fit is not very meaningful if the model itself is almost saturated. 

Minor aspects:

I would not use the label structural equation model if you don’t estimate latent variables. Although path modeling (which I believe is the more appropriate term here) is technically a form of structural equation modeling, it could be misleading given the prominent use of SEM for latent modeling.

Please also report results unstandardized coefficients, standard errors, 95% CIs and beta-coefficients in the text. Each parameter has unique properties. I believe this to be important in order to help the reader in understanding the results and also work with the results (e.g., comparative studies, meta-analyses, etc.)

Please provide descriptive analyses for all conditions (at least mean and standard deviations). Although the overall zero-order correlation table is important too, you should also provide information about each condition. Also: How well did the randomization work? Please provide information about the differences in socio-demographics and other relevant variables.

Please refrain from using the formulation “marginally significant”. In the light of recent discussion around p-values, I would recommend to stick with a priori defined alpha level. 

References

Dienlin, T., & Metzger, M. J. (2016). An extended privacy calculus model for SNSs-Analyzing self-disclosure and self-withdrawal in a U.S. representative sample. Journal of Computer Mediated Communication21, 368–383. 

Rogers, E. M. (2003). Diffusion of innovations (5th ed.). New York: Free Press.

Venkatesh, V., & Davis, F. D. (2000). A theoretical extension of the technology acceptance model: Four longitudinal field studies. Management Science, 46(2), 186–204. 

Venkatesh, V., Morris, M. G., Davis, G. B., & Davis, F. D. (2003). User acceptance of information technology: Toward a unified view. MIS Quarterly, 27(3), 425–478.

Wirth, W., von Pape, T., & Karnowski, V. (2008). An integrative model of mo-bile phone appropriation. Journal of Computer-Mediated Communication, 13(3), 593–617. 

Author Response

Dear Editor, dear Reviewers,

Before we give a detailed response to the reviewers’ comments and report on the changes we have made to the manuscript, we want to thank you for the helpful suggestions and comments on our paper. Please let us emphasize that we really appreciate the high level of constructiveness that characterized all your suggestions. We tried to take up all feedback and feel that the manuscript indeed improved considerably. Most importantly, with the new version of the manuscript, we tried to offer a more fine-grained discussion of our findings and their limitations as well as we also tried to be more explicit about how our study contributes to the present state of knowledge. Based on the feedback of the reviewers we recalculated the scales and the path model resulting in a better model fit. The results can now be interpreted as an even better indication for the weighing of risks and benefits in the privacy calculus model. Below you find an outline of the revisions we have made to the manuscript in accordance with your comments.

Reviewer 2: 

I do appreciate that the authors aimed at recruiting a large sample. Given that much prior work is based on under-powered studies, it is refreshing to read a study that was able to recruit a comparatively large sample. In general, I thus would deem the paper suitable for the audience of Informatics as well as the broader community of privacy and technology scholars.

Authors’ answer:

Thank you so much. We are really pleased that you appreciate the size of the sample.

Reviewer 2:

I also would like to note that I think that trying to understand these processes (technology/application/SNS acceptance/adoption processes) with vignettes and online surveys is of limited value. The major limitations - the hypothetical scenarios and the hypothetical acceptance - that were also acknowledged by the authors are problematic: We need to study these processes under real circumstances where we - as scholars - can observe people’s use of technology and reactions to certain manipulations. Prior studies have shown that intentions to not necessarily align with behaviors and confrontation with actual technology leads to different reactions that hypothetical scenarios.

Authors’ answer:

You are right that hypothetical scenarios can only reflect on real situations and reactions to a limited extent. However, vignette studies represent a common and established methodological approach, especially when investigating the attitude and reactions of individuals to new and expensive technology before implementing it. We absolutely agree that processes need to be studied under real circumstances. Nevertheless, we believe that our work – as a first step – makes an important contribution to research in the field of technology acceptance by showing potential predictors of and tendencies regarding system acceptance and building the basis for further studies where employees are actually confronted with IoT technology. In order to present realistic versions of this technology, for the scenarios of our study we collected information from manufacturer websites (e.g., www.rhebo.com) as well as from existing engineering research (Virgona, Kirchner, & Alempijevic, 2015). To further enhance the representativeness of the results, only employees were addressed, as this group is best able to put themselves into the respective situation. However, we take up your criticism by incorporating a hint to future less hypothetical studies and designs(see p. 13):

Such scenarios are functional by allowing to draw first conclusions as well as making comparisons between different conditions. In order for the evaluation of these scenarios to be realistic, the vignettes of this study describe existing features of smart monitoring systems and provide employees with information they can access due to data protecting legislation, such as the GDPR. However, employees might react differently if their employer actually makes the decision to deploy such a system. For this reason, in further studies, research needs to reflect on employees’ reactions and acceptance processes under real circumstances. 

Reviewer 2:

First, although the authors study technology acceptance, there is no reference to any of the established and well-studied theories that actually aim at predicting technology adoption/acceptance (e.g., TAM by Venkatesh et al.; diffusion theory by Rogers, or even the mobile phone appropriation model by Wirth et al.). This is unfortunate as these theories could have provided more insights into actual acceptance processes and probably would even have suggested other potential predictors/antecedents of technology acceptance. This is a major flaw in the theoretical conceptualization of this study. The authors should at least discuss these approaches and situate there research accordingly, even if this would mean addressing potential neglects in the limitation section. 

Authors’ answer:

We thank the reviewer for the helpful remark. This comment helped us to substantially improve the theoretical part of the manuscript. We agree with the suggestion of the reviewer that considerations regarding technology acceptance models such as TAM (Davis, 1989) and diffusion of innovations theory (Rogers, 2003) should be included to the paper. We therefore added the following section into the literature review (see p. 3):

With regard to technology acceptance, many theories have been proposed by scholars. The technology acceptance model (TAM; Davis, 1989) found substantial and empirical support and is well-studied in this field of research. Particularly, it aims at predicting a person’s behavioral intention to use a system as well as actual usage depending on perceived usefulness (i.e. to what extent the system helps enhancing individual’s job performance) and perceived ease of use (i.e. the required effort of the usage). A theoretical advancement of TAM and other constructs is the unified theory of acceptance and use (UTAUT; Venkatesh, Morris, Davis, & Davis, 2003). This extensive model takes additional factors, such as social influence and voluntariness of use as determinants of usage intention and behavior into consideration. At this point, it becomes clear that theories and models of technology acceptance have a common goal, which is to explain behavioral intention and usage adoption. However, traditional and advanced approaches of technology acceptance do not seem to be entirely applicable in the context under investigation here: Employees are affected by the system but apart from being monitored, they do not interact with it or actively use it. Moreover, they do not have the choice to adopt or reject the system but can only evaluate to what degree they approve the authority decision regarding the system’s implementation at the workplace. Accordingly, most constructs taken up by technology acceptance models, such as investigating the effect of system usage on job performance, are not applicable in this context. Further, in his diffusion of innovations theory Rogers (2003) describes innovation diffusion as an adaptation process where individuals first learn about the innovation’s existence, form an attitude towards it and engage with it before finally putting it into use. However, he points out, that specifically in the context of organizations, an “authority innovation-decision is one with which the organization’s employees must comply” (p. 403). In other words, in the case of the implementation of an IoT-monitoring system at the workplace, employees have no decisional power, which is why common technology acceptance models seem less suitable than the risk-benefit trade-off postulated by the privacy calculus. Last but not least, these models do not consider the privacy aspect, which, given the potentially privacy intrusive character of tracking systems, plays a decisive role for the acceptance of such systems. In the light of the above, privacy calculus serves as the main theoretical foundation investigating system acceptance as an outcome of a risk-benefit trade-off. 

Reviewer 2:

The formulation of almost all hypotheses is imprecise and misleading given the design of the study: All hypotheses that refer to relationships between variables that could be interpreted as causal effects (based on the experimental design; e.g., H3a and H3b) are formulated “relationships". Hypotheses for relationships that cannot be interpreted causally based on the design, however, are formulated as actually effects (e.g., H1, H5). Furthermore, H5b and H6 are exactly the same.

Authors’ answer:

We apologize for the misleading formulation of the hypotheses. H1, H2 and H5 were reformulated as hypotheses testing relationships whereas H3a and H3b were rephrased to investigating causal effects. We also adjusted H6, which was inserted incorrectly.

Hypothesis 1 (H1): There will be a positive relationship between employees’ trust in the company and their acceptance of a smart emergency detection system.

Hypothesis 2 (H2): There will be a negative relationship between trust and perceived risks.

Hypothesis 3a (H3a): The amount of tracking will have a negative effect on employees’ acceptance of the system.

Hypothesis 3b (H3b): The amount of tracking will have a positive effect on perceived risks.

Hypothesis 4 (H4): The rescue value of the smart emergency detection system will have a positive effect on employees’ acceptance of the system. 

Hypothesis 5a (H5a): There will be a positive relationship between perceived benefits and the acceptance of a smart emergency detection system.

Hypothesis 5b (H5b): There will be a negative relationship between perceived risks and the acceptance of a smart emergency detection system.

Hypothesis 6 (H6): Perceived benefits will be negatively related to perceived risks.

Reviewer 2:

Moderation hypotheses that do not specify a direction cannot be tested. Please reformulate to include the direction of the moderating effect.

Authors’ answer:

We are sorry for the wrong formulation of the moderation hypotheses. As you suggested, we included the direction of the moderation into the formulation.

Hypothesis 7 (H7): The effect of tracking on the acceptance of a smart emergency detection system will be stronger among people with high privacy concerns.

Reviewer 2:

Despite a long literature review and a comprehensive theoretical rationale for all the hypotheses, I am not fully convinced by the proposed model and its relationships. 

First, shouldn’t the rescue value manipulation directly affect the perceived benefits. The measure has even a high compatibility with the manipulation. Why did you not include this path in the model?

Second, why should perceived benefits affect perceived risks? This does not make sense theoretically. They could be correlated, but I don’t think it makes sense to assume a causal relationship here. The indirect effect is hence not logical either. 

Authors’ answer:

As suggested by the reviewer, we tested the impact of the rescue value on perceived benefits. Data show that benefits are not affected by the rescue value of the IoT system. Reasons for the missing impact of the rescue value are discussed in the paper (see p. 11).

The rescue value might be too abstract and theoretical to be perceived as a distinctive benefit of the system. Previous studies investigated more concrete or immediate benefits in the privacy calculus such as the free use of a website [33] or convenience provided by IoT-devices [35]. Thus, the rescue value of a smart monitoring system might be perceived as less present compared to the privacy risks of the system and therefore less relevant in the risks-benefits trade-off. This assumption is substantiated by the fact that in the study benefits of the monitoring system were perceived rather low.

Regarding the correlation of perceived risks and perceived benefits we investigated whether there is a relationship between these constructs in order to additionally test the consistency and reliability of data. Furthermore, the significant relationship demonstrates, that there must be a trade-off regarding risks and benefits supporting the application of privacy calculus in the work-related context. 

Reviewer 2:

I am not convinced by the authors argumentation that privacy concerns should moderate the relationship between amount of tracking and technology acceptance. Did you check whether the amount of tracking affected privacy concerns? Could it be that they are similarly affect as perceived privacy risks? A lot of studies investigating the privacy calculus even assume that risks and concerns are somewhat similar. 

Please note that relationships “after” the mediators or between non-manipulated variables cannot be interpreted causally. Strictly speaking even relationships between manipulations and outcome variables can no longer be interpreted causally if that effect is controlled for other cross-sectional relationships.

Authors’ answer:

We are sorry that the argumentation on the choice of privacy concerns as moderating variables was not comprehensible. We did not test whether the amount of tracking affected privacy concerns due to the fact that privacy concerns were measured as a moderating variable, i.e. a trait variable. Our assumption is additionally based on prior studies investigating privacy concerns as moderating variables (McCole, Ramsey, & Williams, 2010; Tan, Qin, Kim & Hsu, 2012; Yun, Han & Lee, 2013). We added this information into the manuscript and hope that this will clarify your questions (see p. 5). Furthermore, results were interpreted more cautiously.

A number of studies showed that people generally tend to be concerned about their privacy [e.g., 34] and that this tendency is affected by individual’s risk perceptions [35] or might for its part influence risk beliefs [36]. This fact might be traced back to what has been described as a universal and cross-cultural need for privacy [37, 38] leading individuals to strongly valuing their privacy [39]. In their survey, Trepte and Masur [40] demonstrated people’s demand to determine what information should be publicly available and the belief that the protection of privacy should be enshrined in the constitution, which the authors referred to as the value of privacy. However, in their study on the value of privacy, Krasnova, Hildebrand and Guenther [41] found that individuals have a varying degree of subjectively perceived privacy concerns, accordingly attaching a different value to their privacy. This might have an effect on the extent to which individuals perceive the data tracking as a threat to their privacy and thereby moderate the relationship between the amount of tracking and the acceptance of the IoT-system. This corroborates other studies investigating the moderating effect of privacy concerns (McCole, Ramsey, & Williams, 2010; Tan, Qin, Kim & Hsu, 2012; Yun, Han & Lee, 2013). Yun, Han & Lee (2013) demonstrated that the effect of performance expectancy on continuous usage intention of location-based services is stronger when people have low privacy concerns. Tan and colleagues (2012) confirmed privacy concerns as a moderator in the TAM. To be more precisely, they posit that the effect of perceived usefulness and perceived ease of use on users’ intention to continue to use social networking sites varies with different levels of privacy concerns. Considering that people show reluctant attitudes when asked for too much personal information [32, 33], we assume that individuals who strongly value their privacy and have higher privacy concerns will rather avoid IoT-technology with a high amount of tracking in order to protect their privacy. Furthermore, as prior studies showed an interdependence between privacy concerns and perceived risks [35, 36], we expect that the privacy risks people see in the deployment of IoT-technology will be higher when privacy concerns and the value of privacy are high.

Reviewer 2:

In general, I strongly believe in the value of openness and transparency. In light of recent meta-scientific discoveries (e.g., replication crisis, questionable research practices, etc.), I suggest to the authors to evaluate whether they have met the standards of the Peer Reviewers’ Openness Initiative (https://opennessinitiative.org/). I also strongly believe that the public availability of all materials, data, and analysis scripts is important not only for evaluating the study’s contribution and methodological soundness, but also to allow other researchers to reproduce this study's results, replicate it independently, or use it in meta-analyses. As I have no insight into the data, the actual analysis script and cannot reproduce the results based on the data at this moment, I do not feel able to evaluate the methodical soundness of this paper.

Authors’ answer:

We absolutely agree with the reviewer about openness and transparency of data. We are happy to provide the original data set.

Reviewer 2:

Please justify your sample size. Did you compute a priori power analyses? If not, please provide results from a sensitivity analysis. 

Authors’ answer:

We did not compute a priori power analysis. We were uncertain, whether the reviewer However, a post-hoc analysis with a probability level of α = .05 revealed that, due to the large sample size, the observed statistical power is 1.

Reviewer 2:

Please provide information about the scales factorial validity and reliability. Please conduct confirmatory factor analyses and also provide more reasonable reliability estimates than Cronbach’s alpha (e.g., measures of composite reliability such as McDonald’s omega). 

Authors’ answer:

This comment really helped us to improve the quality of the data, thank you so much. We conducted confirmatory factor analyses, which led to higher reliability of the used scales. We furthermore user McDonald’s omega as an additional reliability parameter and provided the average variance extracted (AVE) values.

As a result of the confirmatory factor analysis and the AVE however, we decided to exclude the value of privacy scale. 

Reviewer 2:

It seems that the authors do not report all paths estimated in the model. The current model has only 3 degrees of freedom yet, figure 2 suggest that many more relationships were estimated. Please be comprehensive and report all estimated relationships (including potential correlation between exogenous variables). 

Authors’ answer:

We apologize that we did not provide information on all estimated relationships. We added an additional table to the manuscript including all paths estimated by the model (see  Table 4, p. 9).

Reviewer 2:

Interpreting the model fit is not very meaningful if the model itself is almost saturated. 

Authors’ answer:

The reviewer is right that a manifest model reduces the complexity of a structural equation model with latent variables and therefore does not reach full explanatory power. However, we are convinced that the individual relationship paths are conclusive and that an almost saturated model is not implicitly useless. Since theoretical predictions match the observed differences in parameter estimates, the model seems to be valid.

We leave it up to the reviewer to decide whether the model fit is not sufficiently meaningful and therefore should be removed from the manuscript.

Reviewer 2:

I would not use the label structural equation model if you don’t estimate latent variables. Although path modeling (which I believe is the more appropriate term here) is technically a form of structural equation modeling, it could be misleading given the prominent use of SEM for latent modeling.

Authors’ answer:

Thank you for pointing this out. We refrain from the term structural equation modeling and will use path modeling instead. 

Reviewer 2:

Please also report results unstandardized coefficients, standard errors, 95% CIs and beta-coefficients in the text. Each parameter has unique properties. I believe this to be important in order to help the reader in understanding the results and also work with the results (e.g., comparative studies, meta-analyses, etc.)

Authors’ answer:

We apologize that we did not provide more detailed description of the parameters. We fixed this by reporting unstandardized coefficients, standard errors, 95% CIs and beta-coefficients in the text, as suggested by the reviewer.

Reviewer 2:

Please provide descriptive analyses for all conditions (at least mean and standard deviations). Although the overall zero-order correlation table is important too, you should also provide information about each condition. Also: How well did the randomization work? Please provide information about the differences in socio-demographics and other relevant variables.

Authors’ answer:

We added further information into the manuscript including descriptive analyses for all conditions, information regarding the randomization as well as differences among conditions (see Appendix A, p. 14):

Reviewer 2:

Please refrain from using the formulation “marginally significant”. In the light of recent discussion around p-values, I would recommend to stick with a priori defined alpha level.

Authors’ response:

Thank you for this comment. We agree that levels of significance and reliability should be pre-defined and complied with. We fixed that by rephrasing the sentences. 

Thank you again for the numerous insightful comments and suggestions!

Best wishes,

The authors

References

McCole, P., Ramsey, E., & Williams, J. (2010). Trust considerations on attitudes towards online purchasing: The moderating effect of privacy and security concerns. Journal of Business Research63(9-10), 1018-1024.

Tan, X., Qin, L., Kim, Y., & Hsu, J. (2012). Impact of privacy concern in social networking web sites. Internet Research22(2), 211-233.

Virgona, A., Kirchner, N., & Alempijevic, A. (2015, November). Sensing and perception technology to enable real time monitoring of passenger movement behaviours through congested rail stations. In Australasian Transport Research Forum. ATRF.

Wenninger, H., Widjaja, T., Buxmann, P., & Gerlach, J. (2012). Der „Preis des Kostenlosen“. Wirtschaftsinformatik & Management4(6), 12-19.

Yun, H., Han, D., & Lee, C. C. (2013). Understanding the use of location-based service applications: Do privacy concerns matter?. Journal of Electronic Commerce Research14(3), 215.

Zheng, X., Cai, Z., & Li, Y. (2018). Data linkage in smart internet of things systems: A consideration from a privacy perspective. IEEE Communications Magazine56(9), 55-61.

Reviewer 3 Report

I'd like to thank the authors for submitting their paper. I agree with the authors that the user's perception of privacy and trust is an important factor for the acceptability of IoT devices. The investigation of the working context is novel and relevant. The paper is overall well written and easy to understand. The authors' methodological approach is appropriate in general. While I enjoyed reading this paper, I identified some aspects which should be addressed in a revision to further improve the quality of this submission.

I will first address my assessment of specific aspects above and provide further details for the aspects which have not been 100% satisfactory. 

#Does the introduction provide sufficient background and include all relevant references?

I would have expected to learn more about the specific related work in the area of IoT and trust/privacy.

#Is the research design appropriate?

While the research design is appropriate in most aspects, the authors did not investigate the participants' privacy perception of IoT devices in general. This is a serious limitation as the perception of IoT devices is likely to have an impact on the overall perception of privacy in the investigated context.

#Are the conclusions supported by the results?

Since an important factor was not considered in the model (i.e., perception of general IoT privacy), the conclusions are based on an incomplete picture. Still, I would argue that most of the conclusions can be made with an adequate level of confidence.

In addition, I would recommend to address the following issues:

1. The authors considered one specific scenario with a specific IoT System (i.e., emergency monitoring system). Based on the results, the authors make conclusions for IoT devices in general. I wonder how generalisable the tested scenarios really were. There are different IoT systems in the market. Would authors expect different results if they'd named a different system in the scenarios. E.g., a time management system. 

2. Which platform / way of distribution did the authors use to rollout their survey? The high number of female participants indicates a selection bias. Therefore, it would be important to know how the survey was distributed and what compensation the participants received. 

3. The authors main finding is based on the fact that "57% of the variance of the system’s acceptance could be explained by the tracking amount, perceived risks and benefits". While this is not a ground-breaking result, it is nice that the authors can provide evidence-based data. However, I would argue that the authors discussion of the findings is rather shallow and does rather represent a summary of the findings. I wonder if the authors could provide further thoughts on the implications for practitioners, developers, and researchers. What is the actual take-away message of this finding? The implication for employers "to implement privacy-by-design when deploying smart technology capable of data tracking in order to ensure acceptance of this technology" is not specific enough. In large parts of the world, this is already enforced by law.

Author Response

Dear Editor, dear Reviewers,

Before we give a detailed response to the reviewers’ comments and report on the changes we have made to the manuscript, we want to thank you for the helpful suggestions and comments on our paper. Please let us emphasize that we really appreciate the high level of constructiveness that characterized all your suggestions. We tried to take up all feedback and feel that the manuscript indeed improved considerably. Most importantly, with the new version of the manuscript, we tried to offer a more fine-grained discussion of our findings and their limitations as well as we also tried to be more explicit about how our study contributes to the present state of knowledge. Based on the feedback of the reviewers we recalculated the scales and the path model resulting in a better model fit. The results can now be interpreted as an even better indication for the weighing of risks and benefits in the privacy calculus model. Below you find an outline of the revisions we have made to the manuscript in accordance with your comments.

Reviewer 3: 

I'd like to thank the authors for submitting their paper. I agree with the authors that the user's perception of privacy and trust is an important factor for the acceptability of IoT devices. The investigation of the working context is novel and relevant. The paper is overall well written and easy to understand. The authors' methodological approach is appropriate in general. While I enjoyed reading this paper, I identified some aspects which should be addressed in a revision to further improve the quality of this submission.

Authors’ answer:

Thank you so much. We are really pleased that you enjoyed reading our manuscript! We will be glad to address your suggestions in order to improve our submission.

Reviewer 3:

I would have expected to learn more about the specific related work in the area of IoT and trust/privacy.

Authors’ answer:

We apologize for the lack of literature investigating the concepts of trust and privacy in the context of IoT technology. Following your comment, we added further information in to the manuscript, discussing the impact of legal requirements on the implementation of a monitoring system at the workplace, research addressing privacy preserving measures for IoT systems and further findings on trust (see pp. 2-3):

In their study on the intention to use wearable devices at the workplace, Yildirim and Ali-Eldin (2018) demonstrated that employees with a high level of privacy concerns in terms of data collection and improper access showed little trust in the employer.

In order to raise an organization’s efficiency, workplace monitoring has become a convenient instrument for employers, promising positive outcomes such as a higher security of employees (e.g., by identifying potential safety hazards) or support of HR decision-making by providing insights about job performance. Recent literature provides an overview of different approaches regarding the development of privacy-preserving IoT technology. Studies demonstrate technical possibilities to reduce privacy risks, for example by providing control over data to be collected (Huang, Fu, Chen, Zhang, & Roscoe, 2012), authorization restriction (Alcaide, Palomar, Montero-Castillo, & Ribagorda, 2013) or continuous anonymization (Cao, Carminati, Ferrari, & Tan, 2011). However, numerous IoT systems still have no privacy protecting measures leading to enormous challenges for employee privacy as a result of the vast collection of personal data. Intelligent IoT tracking systems are able to gather masses of sensitive data correlating it to employee performance and automatically drawing conclusions on improvable aspects regarding efficiency, which is called people analytics (Moore, Upchurch, & Whittaker, 2017). From the legal point of view, employees give up a lot of privacy expectations, since “The employer is allowed to monitor employees through supervisors, video cameras, computer software, or other methods that capture employees working within the scope of employment.” (Bodie, Cherry, McCormick, & Tang, 2017, p. 26). However, there are limits to employee surveillance, for example if monitoring is undisclosed. Secret tracking requires severe and legitimate reasons, such as suspicion of fraud or theft (Bodie et al., 2017). Generally, employees should be provided with information about being subject of data collection (e.g., by information signs). Accordingly, the commencement of the European General Data Protection Regulation (GDPR) guarantees individuals the right to be informed about the processing purposes, the legal basis or automated decision-making processes. It should be noted, however, that legal regulations between countries have to be differentiated (Atkinson, 2018).

Reviewer 3:

While the research design is appropriate in most aspects, the authors did not investigate the participants' privacy perception of IoT devices in general. This is a serious limitation as the perception of IoT devices is likely to have an impact on the overall perception of privacy in the investigated context. 

Authors’ answer:

You are right that the general perception regarding the privacy of IoT technology might influence attitude and behavior regarding these devices as well as the perception of potential privacy threat when implementing such systems. However, due to the high level of heterogeneity of IoT devices and the diverse fields of their implementation, an assessment of the perception of this technology in general proves to be rather difficult. In terms of privacy, an individual might distinguish between technology used at home versus technology used by the employer. Furthermore the evaluation of IoT systems in regard of potential privacy threat might depend on the respective device. It is conceivable that a smart freezer will raise less privacy concerns than for example an intelligent speaker, like Amazon’s Alexa. There are approaches to investigate users’ perception of IoT systems (Economides, 2016). However, it is stressed that the perception of a particular system depends on a variety of factors, such as user characteristics (age, competencies) or IoT system’s characteristics (usability, security or brand name) thus requiring an individual assessment. In order to acknowledge your comment, we included this specification into the method section.

Reviewer 3:

Since an important factor was not considered in the model (i.e., perception of general IoT privacy), the conclusions are based on an incomplete picture. Still, I would argue that most of the conclusions can be made with an adequate level of confidence.

Authors’ answer:

Thank you very much for your approval. We appreciate that you nevertheless find our work to be reliable.

Reviewer 3:

The authors considered one specific scenario with a specific IoT System (i.e., emergency monitoring system). Based on the results, the authors make conclusions for IoT devices in general. I wonder how generalisable the tested scenarios really were. There are different IoT systems in the market. Would authors expect different results if they'd named a different system in the scenarios. E.g., a time management system.  

Authors’ answer:

We absolutely agree that results from scenario testing are only generalizable to a limited extent. Still, vignette studies represent a common and established methodological approach, especially when investigating the attitude and reactions of individuals to new and expensive technology before implementing it. Therefore, we believe that conclusions drawn from the results are applicable with regard to monitoring IoT systems at the workplace. Nevertheless, other IoT devices might be evaluated differently in terms of perceived privacy, risks and benefits. Therefore, we take up your criticism by incorporating a hint that IoT systems need to be investigated with respect to specific characteristics (see p. 13):

Caution should also be taken with regard to generalizability of IoT system acceptance. Due to the heterogeneity and various characteristics of IoT devices, IoT systems might be evaluated differently in terms of privacy, perceived risks and benefits.

Reviewer 3:

Which platform / way of distribution did the authors use to rollout their survey? The high number of female participants indicates a selection bias. Therefore, it would be important to know how the survey was distributed and what compensation the participants received.

Authors’ answer:

We added further information in to the manuscript that will hopefully clarify your questions (see p. 6):

As the focus of the study is on electronic monitoring at the workplace, the study was directed at employees in order to include participants who are familiar with the context. Participants were recruited online via Facebook groups and survey distribution platforms (e.g., www.soscisurvey.de) and had the chance to win a gift card worth 5-100€. A total of 770 participants took part in the online-survey. In a first step, we filtered out participants who were underage, failed to respond to the control questions and those who showed unreasonable reading times, which we tested to be less that 220 seconds, resulting in an overall sample size of 661 individuals. The sample included 206 males, 447 females and 8 individuals who did not specify their gender with an age range of 18 to 99 (M= 27.78, SD= 7.99). 

Regarding the high number of female participants, we agree that a balanced sample increases the representativeness of the study and therefore acknowledged the gender ratio as a limitation of our study. Unfortunately, the imbalance between the number of men and women participating in psychological online studies is not unusual (e.g., Meier & Neubaum, 2019). Gender differences regarding privacy and the acceptance of smart technology were not to be expected (Krasnova, Veltri & Günther, 2012; Malhotra, Kim, & Agarwal, 2004). However, we conducted an independent-samples t-test to compare gender-related system acceptance.There was no significant difference in men (= 4.68, SD = 1.56) and women (= 4.62, SD = 1.67); t(651) = -.47, p= 0.636.

Reviewer 3:

The authors main finding is based on the fact that "57% of the variance of the system’s acceptance could be explained by the tracking amount, perceived risks and benefits". While this is not a ground-breaking result, it is nice that the authors can provide evidence-based data. However, I would argue that the authors discussion of the findings is rather shallow and does rather represent a summary of the findings. I wonder if the authors could provide further thoughts on the implications for practitioners, developers, and researchers. What is the actual take-away message of this finding? The implication for employers "to implement privacy-by-design when deploying smart technology capable of data tracking in order to ensure acceptance of this technology" is not specific enough. In large parts of the world, this is already enforced by law.

Authors’ answer:

We are grateful for this suggestion as it helps adding a practical view to the study. Therefore, as proposed by the reviewer, we extended our discussion significantly by elaborating on the implications for employers (see p. 13). Furthermore, with regard to the recalculated model, we have broadened the reflections on the construct of trust and discussed the risk-benefit trade-off regarding monitoring systems in the specific work-related context in greater detail. 

Implications, which can be drawn for employers, are: first that it is in their interest to protect privacy of staff by limiting employee monitoring or deploying technology with the privacy-by-design approach. In other words, considering the privacy of the workforce before implementing an IoT system enables the employer to resort to a system already working in a privacy-preserving way by its technical implementation, which thus, is more likely to be accepted by the employees. This decision might be a confidence-building measure insuring a responsible handling of employees’ data respecting their value of privacy. Second, if the decision is in favor of installing a system that does not automatically cover the privacy of the employees by virtue of its functioning and thus does not have a privacy-by-design approach, the employer nevertheless has certain possibilities of influencing the acceptance of the system. On the one hand, the employer could provide his employees with detailed information regarding purpose and reasons for collecting the data. On the other hand, the employer could emphasize the benefits, such as security, that the system brings to the employees. Any measures that help to increase the acceptance of the new technology in the company are fundamentally beneficial in order not to jeopardize trust, commitment and performance.

Thank you again for the numerous insightful comments and suggestions!

Best wishes,

The Authors

References

Atkinson, J. (2018). Workplace Monitoring and the Right to Private Life at Work. The Modern Law Review81(4), 688-700.

Bodie, M. T., Cherry, M. A., McCormick, M. L., & Tang, J. (2017). The Law and policy of people analytics. U. Colo. L. Rev.88, 961.

Economides, A. A. (2016, July). User Perceptions of Internet of Things (IoT) Systems. In International Conference on E-Business and Telecommunications (pp. 3-20). Springer, Cham.

Krasnova, H., Veltri, N. F., & Günther, O. (2012). Self-disclosure and privacy calculus on social networking sites: The role of culture. Business & Information Systems Engineering4(3), 127-135.

Malhotra, N. K., Kim, S. S., & Agarwal, J. (2004). Internet users' information privacy concerns (IUIPC): The construct, the scale, and a causal model. Information systems research15(4), 336-355.

Meier, Y., & Neubaum, G. (2019). Gratifying Ambiguity: Psychological Processes Leading to Enjoyment and Appreciation of TV Series with Morally Ambiguous Characters. Mass Communication and Society. DOI: 10.1080/15205436.2019.1614195

Moore, P. V., Upchurch, M., & Whittaker, X. (2018). Humans and machines at work: monitoring, surveillance and automation in contemporary capitalism. In Humans and Machines at Work (pp. 1-16). Palgrave Macmillan, Cham.

Reviewer 4 Report

Summary. The manuscript presents a research in which the participants' willingness in accepting an IoT monitoring system is studied. The paper lists a number of hypotheses through which research questions regarding acceptance of the privacy-preserving vs. privacy-invasive is investigated along with the rescue factor. 

Evaluation. The manuscript is very well-written and also interesting. It explores the acceptance of monitoring technology in comparison with privacy concerns. 

There are a few comments that seems, if addressed, will enhance the quality of the work:

1) I could not see whether the "effect size" has been measured for each factor. Please high light it, if it is, otherwise, provide it.

2) Figure 2, there is a broken link between "privacy concerns" and "Trust in the employer"

3) Regarding "Trust in the employer" it is not clear whether the significant relation between trust and acceptance of IoT is due to "Trust" or "domination" of the employer. The participants will accept the technology if they trust their employer or they have to accept the technology no matter what?

4) The descriptions given on Page 9 (lines 5 - 29) provide some information whether there is a significant relationship between factors or not, but it does not provide any reason or why it is the case.

5) The Appendix Z only provides 4 scenarios, but it does not clearly describes how the factors are controlled.

Round 2

Reviewer 1 Report

The authors have carefully responded to each of my comments through added sections or greater discussion of specific sections.  The best addition involves the relatively short implications for practitioners.  It needs to be short given the reduced ability to generalize the results of the study due to the "ideal" conditions presented.

Overall, I have qualms about the paper due to its reduced relevance to the real workplace due to a messy world,  Unfortunately--

Employees might not actually know about the system's privacy policies and protections. The European Union and various international privacy laws may authorize providing various privacy policy documents, but many of the documents are hard to read and may cloak what is really going on. e.g. Cranor et al (2015)  Are they worth reading?  An in-depth analysis of online trackers' privacy policies.  I/S:  A journal of Law and Policy for the Information Society, 11, 325-404; Rowan and Dehlinger (2014)  A privacy policy comparison of health and fitness-related mobile applications.  Procedia Computer Science 37, 348-355.

Employees might not care about the system's privacy policies and protections.

Employees might not be aware of all the privacy-reducing technology around them.

I do not know how to fix this reduced relevance due to the compromises that were necessary in the design of the experiment.

The main change would be adding to the paragraph that states commencement of the European General Data Protection Regulation (GDPR) guarantees...."   Even though it guarantees individuals' right to be informed, it does not guarantee they are actually informed.

In the future literature section, these caring and awareness concerns should be mentioned.

On error on page 12 is a simple one..."On the one hand, the employer could provide HIS employees with..."

Reviewer 3 Report

I'd like to thank the authors for submitting a revised version of their paper. It is great to see that the paper significantly improved and that the reviews were helpful. Based on the revised version of the paper which sufficiently addresses my previous concerns, I will recommend acceptance.

Author Response

Dear reviewer,

we are very glad about your decision to recommend acceptance. We would once again like to thank you for the valuable suggestions which helped to significantly improve our paper. 

Kind regards